# Gigantic jet discharges evolve stepwise through the middle atmosphere

Oscar A. van der Velde [1], Joan Montanyà[1], Jesús A. López[1] & Steven A. Cummer[2]

In 2002 it was discovered that a lightning discharge can rise out of the top of tropical thunderstorms and branch out spectacularly to the base of the ionosphere at 90 km altitude. Several dozens of such gigantic jets have been recorded or photographed since, but eluded capture by high-speed video cameras. Here we report on 4 gigantic jets recorded in Colombia at a temporal resolution of 200 μs to 1 ms. During the rising stage, one or more luminous steps are revealed at 32-40 km, before a continuous final jump of negative streamers to the ionosphere, starting in a bidirectional (bipolar) fashion. The subsequent trailing jet extends upward from the jump onset, with a current density well below that of lightning leaders. Magnetic field signals tracking the charge transfer and optical Geostationary Lightning Mapper data are now matched unambiguously to the precisely timed final jump process in a gigantic jet.

[1] Lightning Research Group, Electrical Engineering Department, Universitat Politècnica de Catalunya – BarcelonaTech, Colon 1, Terrassa 08222, Spain. [2] Electrical and Computer Engineering Department, Duke University, PO Box 90291, Durham, NC 27708, USA. Correspondence and requests for materials should be addressed to O.A.v.d.V. (email: oscar.van.der.velde@upc.edu)

While the cousins of gigantic jets in the mesosphere, red sprites, have been studied with high speed cameras for a long time[1–4], blue and gigantic jets, due to their rarity, still need to be unraveled long after their discovery. Gigantic jets develop out of lightning leaders into a more sprite-like discharge. Electrostatic and thermodynamic models have explained some features of the initial development of gigantic jets[5–9]. Sprites and other discharges like Saint Elmo's Fire consist of cooler ionized filaments (<500 K) called streamers, left behind by propagating waves of electron avalanches. In contrast, lightning channels are hot (>5000 K) conductive plasma channels that polarize and self-propagate by focusing the electric field around their tips[6,10,11]. These propagating tips of lightning channels are called leaders.

In the tropics gigantic jets typically transfer negative charge up to the ionosphere[12–20]. It has been demonstrated that unbalanced thunderstorm charge structures allow lightning leaders to escape from the cloud in the form of different types of cloud-to-ground flashes, as well as gigantic jets[21]. Overshooting convective cloud tops may help create a charge structure which makes the channel grow more vertically out of the cloud[22]. The lightning flash itself may first neutralize a positively charged part of the cloud, helping create a stronger imbalance triggering the jet[16]. Such requirements apparently are uncommon in nature given the rarity of gigantic jets.

The fan-shaped structure in gigantic jets above 40 km altitude is generally accepted to consist of streamers like those in carrot sprites. It is commonly believed[8,11,12,15,18,23,24] that the leader from the cloud reaches up to this altitude owing to the similarity in morphology of one or two well-defined narrow channels and its upward speed similar to lightning leaders ($10^4$–$10^5$ m s$^{-1}$).

On the other hand, it is quite regularly observed[13,16,17,19,24] that gigantic jets complete the cloud to jump altitude already with speeds of more than $10^6$ m s$^{-1}$. In addition, some events show streamer-like morphology already from the cloud top: abundant upward filaments, irregularities in brightness, intertwined look[13,17], and typical V-shaped branching angles as observed in laboratory streamers[25,26]. The lower jet displays a blue/purple color in photographs[17,24,27,28], with gradually less blue and more red above 35 km[17,24]. Such color shift was also observed in large sprites[29] and is attributed to quenching of the $N_2$ first positive and negative band emissions[30]. The lower jet expands laterally with time[17,23,31] and sometimes side branches connect again to the completed main channel[17] as is observed in late streamers in the laboratory[32] and in sprites[2,4]. Such reconnections have not been observed in normal lightning leaders.

A brightness transition has been identified[33] in the lowest section in the gigantic jet images of Pasko et al.[12] where the thermal streamer-to-leader transition is taking place. Their supplementary images[12] remarkably show how the lowest sections of two separate jet branches (frame 9) later fused into one bright stem (frame 12 and after). Similar persistent bright stems can be found between 18 and 26 km in the video images of close range recent jet events[16,18,24,34] resembling cloud-to-air lightning branches and their streamer coronas[35]. This stem flickers simultaneously with the cloud lightning flash, and often reappears brightest at the end of the event[11,12,16–18,36]. Given the different interpretations of the leader/streamer nature of the lower half of the jet, more detailed recordings of gigantic jet dynamics are desired.

Here we report the evolution of gigantic jets observed at high imaging rates (0.2–1.1 ms) in Colombia. We find that the final jump to the ionosphere starts at ~35–40 km altitude in a bidirectional (bipolar) way. Before reaching this level, two events display one or more 2–5 km size steps which do not exhibit the behavior typical of stepped leaders in lightning. The jump causes a sharp rise of current moment, followed by a minimum of several ms before the main charge transfer and luminosity pick up during the initial fast rising stage of the top of the trailing jet, whose origin can be traced back to the onset altitude of the final jump.

## Results

**Observation campaigns in Colombia**. Two campaigns were conducted at the north coast of Colombia, in summer 2017 and fall 2018. The first campaign used a portable intensified fast camera system, operated near Santa Marta city at 900 images per second. The second campaign was conducted from Barranquilla and Cartagena and fielded a faster system operated at 5000 images per second. The two campaigns spanned 3 months and resulted in 12 gigantic jets in 6 different nights, 5 of which recorded successfully by the high speed camera (see Supplementary Movies 1 and 2). The events occurred at distances of 318 (GJ 12), 354 (GJ 3), 346 (GJ 4), and 370 km (GJ 6 and 7) from the observer, and were usually located by lightning detections along the observed azimuth (Supplementary Fig. 1). Altitudes of gigantic jet features have been determined (±1 km precision) based on elevation above the horizon after fitting the images to the star background, as explained in more detail in the Methods section.

A meteorological study of the producing storms is outside the scope of this study, but satellite images (Supplementary Fig. 2) reveal rapidly expanding, cold cloud tops at the time of the jets (−85 and −90 °C for GJ 7 and GJ 12, reaching up to 16.5 km). In the case of 14 August 2017 (GJ 3 and 4), the tops were relatively warm at −65 °C, corresponding to 14.5 km above sea level. Of interest is also that a negative sprite was recorded 2 min after GJ 4, and 18 min after GJ 12 (Supplementary Fig. 3). Such association has been noted before[37].

**Morphological features and variety among gigantic jets**. Figure 1 shows the images of 6 gigantic jet events. Morphological differences between the events are notable. GJ 4 is the archetypical tree-like gigantic jet[13] while GJ 12 resembles the carrot-like gigantic jet[13] with a more massive central structure, beads and patches at the top, like those seen in carrot sprites. GJ 7 has two clearly separated main branches, which did not develop simultaneously, as shown later. GJ 2 reached only 67 km altitude, appearing like a trailing jet only, without developing the branched out top of most gigantic jets. Its evolution is shown in more detail in Supplementary Fig. 4. GJ 3, GJ 4, GJ 7, and GJ 12 respectively reached 86.1 km, 89.5 km, 89.6 km, and 89.6 km above sea level. It can be noted that GJ 2, GJ 4, and GJ 12 as well as the two branches of GJ 7 are slightly tilted from the vertical. GJ 6 was a very short-lasting, highly branched event with minimal trailing jet features and interesting morphology. Unfortunately, that event (like GJ 2) was not successfully recorded by the high speed camera. The lower part was not visible due to clouds. Events 8–11 of 2 November 2018 are not shown. Only their tops above 60 km were visible between cloud cover, but had unusual characteristics to distinguish them from sprites.

Figure 1b indicates the Leading Jet (LJ), Fully Developed Jet (FDJ), and the Trailing Jet (TJ) stages of event GJ 3 as recorded by the low speed but higher resolution camera. The LJ is just above the perception threshold, owing to the long distance through dense atmosphere near the horizon. It lasts ~150 ms. GJ 3 and GJ 4 are the only recorded events of the series where the lowest parts of the jet are clearly visible, starting from 20.0 km (GJ 4) and 20.4 km (GJ 3), but the extensive anvil cloud of the storm itself masks the true exit point at the cloud top. These two cases revealed a bright final cloud leader after the end of the TJ. These

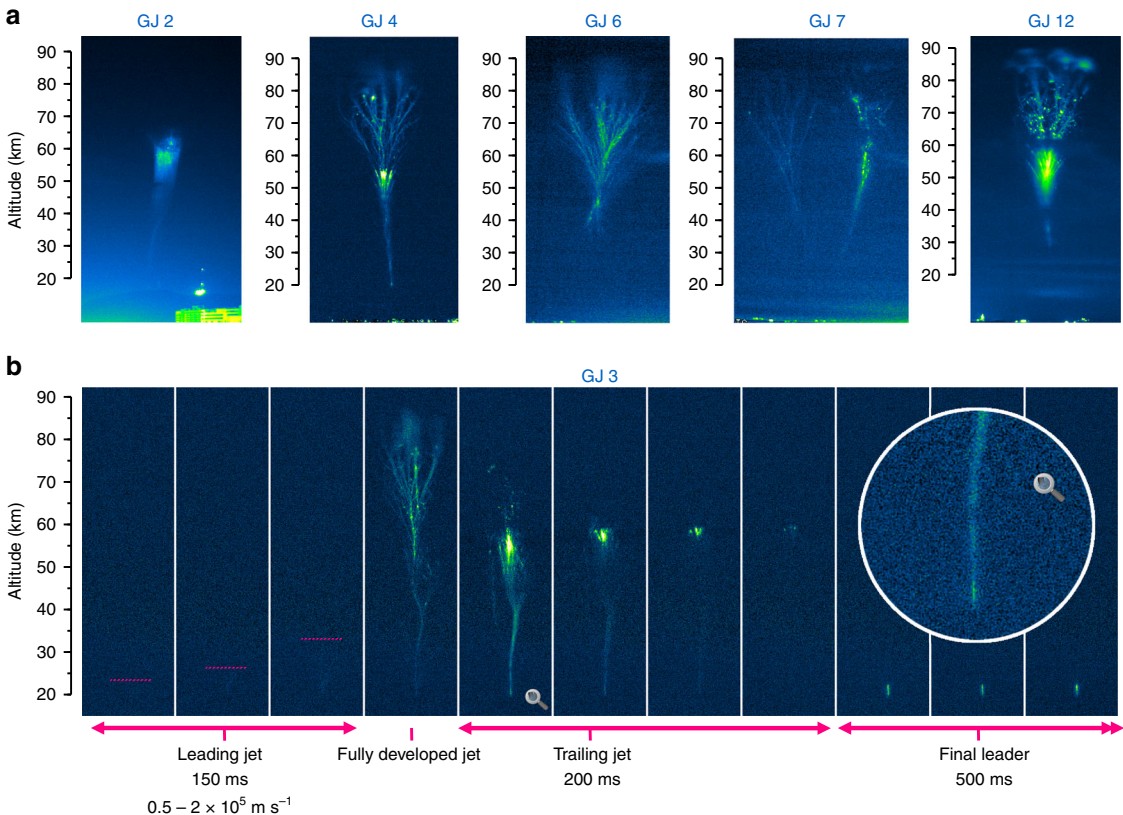

**Fig. 1** Five gigantic jets captured from the north coast of Colombia. **a** Jets recorded from Santa Marta (GJ 2-4), Barranquilla (GJ 7), and Cartagena (GJ 12). **b** The Leading Jet, Fully Developed Jet, Trailing Jet, and the Final Leader stages are shown for the case of GJ 3 recorded by the 2.3-megapixel camera at 20 images per second. The uneven background was subtracted, horizontal banding noise removed, and contrast enhanced. A close-up of the lower jet and cloud leader section one frame after the FDJ is included in the circular insert

reached up to 22.9 and 22.0 km altitude, lasting 500 ms and 17 ms for GJ 3 and 4, respectively. This section does not appear out of nowhere: one frame after the FDJ a section of the jet near the visual cloud top of only one pixel wide and 16 pixels tall (GJ 3) remains notably brighter, topping at 21.7 km and 20.5 km respectively. In GJ 4 a similar bright section is moving upward (Supplementary Fig. 5). Like the events of Soula et al.[17], the lower jet channel broadens significantly, to 500–1000 m width measured at 30 km altitude shortly after the FDJ.

**General evolution of gigantic jets in high speed camera sequences.** Figures 2 and 3 show the image series of 4 gigantic jets captured by the intensified high speed camera systems. Figures 4 and 5 show the evolution in the form of time-altitude-brightness plots. In these plots the fast vertical developments show as narrow, almost vertical lines, while persistent channel luminosity and beads show as horizontal streaks. The LJ, FDJ, and TJ stages are indicated for GJ 3 in Fig. 4a and especially the latter two are easily recognized in all events.

GJ 3 and 4 appear very similar in evolution. GJ 3 and 12 display briefly luminous segments (steps) during the late LJ phase around 35 km altitude, which are missing in GJ 4. The significantly brighter GJ 12 features a renewed glow of old FDJ beads above the TJ channel, with simultaneous crown-like patches at the top of the jet. The multiple-branched GJ 7 looks very different at first sight. On closer inspection, it turns out that there are 3 FDJ developments and 3 corresponding TJ. Most of the complexity arises in the top region of the first attempted FDJ. The details will now be discussed by evolution stage.

**Leading Jet stage.** While the detection camera could barely make out the LJ starting ~150 ms before the FDJ (GJ 3 and 4), in none of the events a continuous upward development of a filament is detected in the high-speed camera images. Before the final jump in GJ 3, three segments (or steps) became visible at 32.2–35 km, 35.2–40.0 km, and 40.0–45.1 km altitude (marked by numbers 1, 2, and 3 in Fig. 2a), with two silent intervals of 10 ms between them. As a possible last step (marked 4), a bright forked structure starts at 50.3 km altitude, following immediately (<1.1 ms) after segment 3 in which it appears to be rooted. While initially barely visible above the background noise, these small LJ segments are part of the channel structure visible in the FDJ frames, verified by stacking the images of the leading jet stage. They also align stepwise in the time-altitude plot (Fig. 3a). It can be observed that the brightness of the segments increases with altitude, or perhaps by segment size or speed. If there were any continuously luminous filaments, they were not detectable in the fast camera images. The speed of upward extension during the first three steps of GJ 3 is faster than $5 \times 10^6$ m s$^{-1}$ (considering the duration may well be shorter than 1.1 ms), which is about two orders of magnitude faster than the time-averaged speed below 30 km during the LJ stage ($5 \times 10^4$ to $2 \times 10^5$ m s$^{-1}$) observed with the slow camera. The time-averaged propagation speed between 32 and 40 km is $6 \times 10^5$ m s$^{-1}$.

Similarly, GJ 12 exhibits a clearly recorded segment 5.0 ms before the start of the final streamer development to the FDJ, at an altitude of 32.6 km to 34.4 km. The lowest visible altitude during the TJ stage is 28.5 km but no features could be discerned below 32 km during the LJ. An inserted close-up image in Fig. 2c

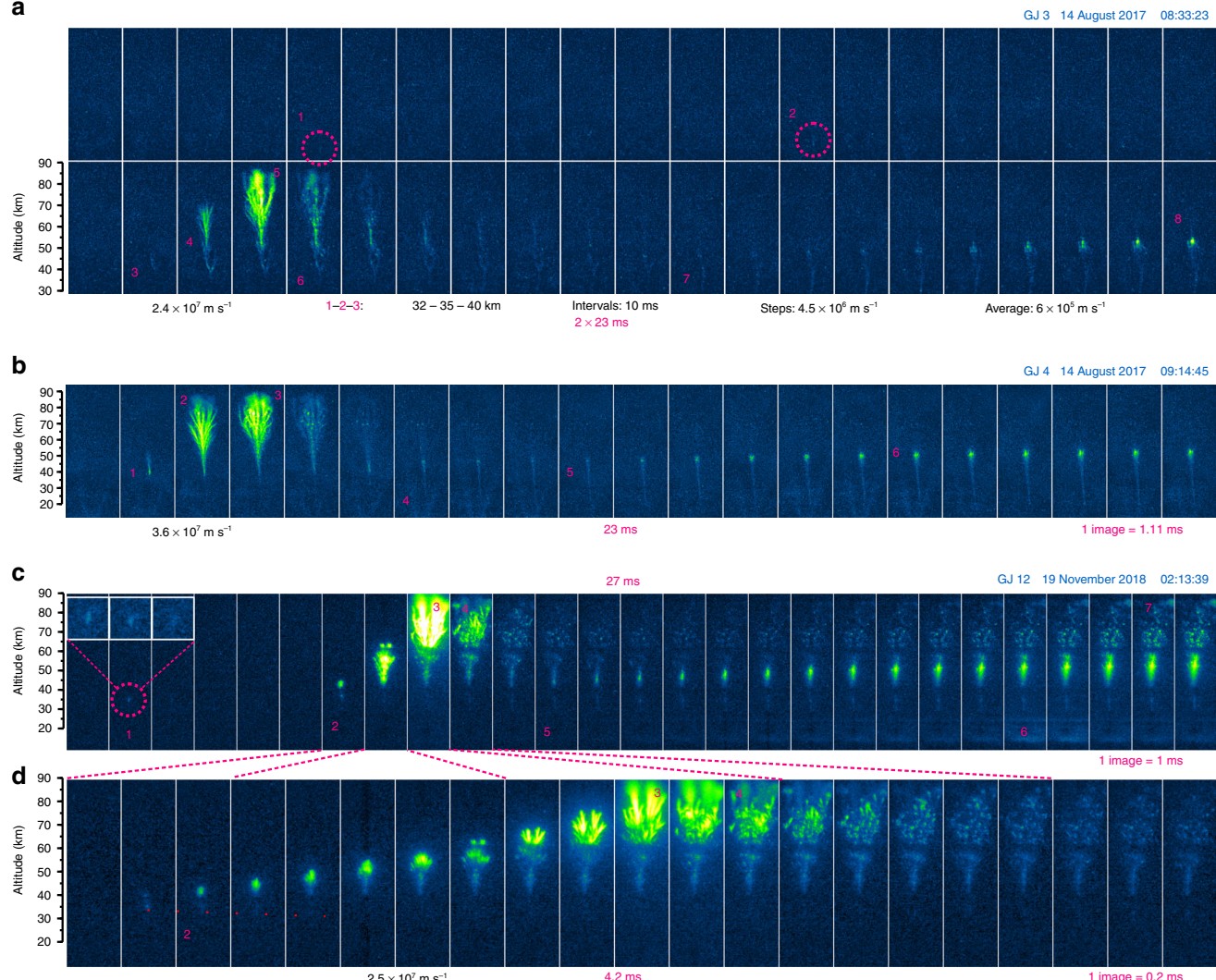

**Fig. 2** Development of gigantic jets 3, 4, and 12 as recorded by the high speed camera. The selected image sequences focus on the upward development and the onset of the trailing jet in **a** GJ 3, **b** GJ 4, and **c** GJ 12 at ~1 ms resolution. Panel **d** is a zoom in time of the FDJ stage of GJ 12 with a temporal resolution of 0.2 ms. A dotted line indicates the brief downward extension of a streamer starting at the same time as the final upward jump. An insert in panel **c** shows the step feature in 0.2 ms resolution. Numbered features are described in the text. The images had their background sky gradient subtracted, but in **b** transient V-shaped patterns caused by a human light source near the image bottom remain visible

shows the three subsequent 0.2 ms frames of the step segment. During the second and third frame the segment occurs lower than in the first frame, which give it a slightly skewed look in the magnified time-altitude-brightness section (Fig. 5a). This could indicate a possible bidirectional extension, or a brief upward extension followed by brightening at the root. The upward speed corresponds with $>9 \times 10^6$ m s$^{-1}$. As shown by luminosity curves in Fig. 6, the mean luminosity trace C at the altitude of the steps (28–38 km) increased slightly before the segment was observed and maintained this level also after the segment disappeared, until the final FDJ development.

**Fully Developed Jet stage**. The start of the final growth of the jet to the ionosphere is brighter than the LJ and easily identified in the images. From this point an upward diverging branched structure develops without interruption. The onset of this final jump is found at 50.3 km (GJ 3), 38.4 km (GJ 4), 41.3 km (GJ 7), and 35.3 km (GJ 12) above mean sea level. On the other hand, GJ 7 is an exception in continuity as it stopped at 68 km, requiring a

second development to make it to 77 km altitude, forming lasting beads or bright segments (Fig. 5b). GJ 2 of Fig. 1a never continued beyond approximately the same height, 68 km. GJ 3 also behaves differently in the sense that the onset of (what seems) continuous upward growth started from 40.0 km but the diverging, brightening branches morphologically similar to the others did not occur until above 50.3 km.

The onset of the jump in GJ 7 and GJ 12 is observed to be accompanied by a downward extending filament which fades after a few frames. In GJ 7, this bidirectional development (Fig. 3a) clearly originates from a bright bead (1), which remains luminous for 1.4 ms. The negative streamer heads (2) propagate away without much brightness in their wake, while the positive end is luminous with a modest visible extension down to 38.7 km. At (3) this section re-illuminates briefly. Note that later during the event the jet channel was detectable down to 32.5 km altitude in normal-speed images.

In GJ 12 (Fig. 2d), the origin appears a brighter spot just below a cloud band, but the downward filament can be distinguished by a slope in Fig. 5a, extending down to the altitude from which the

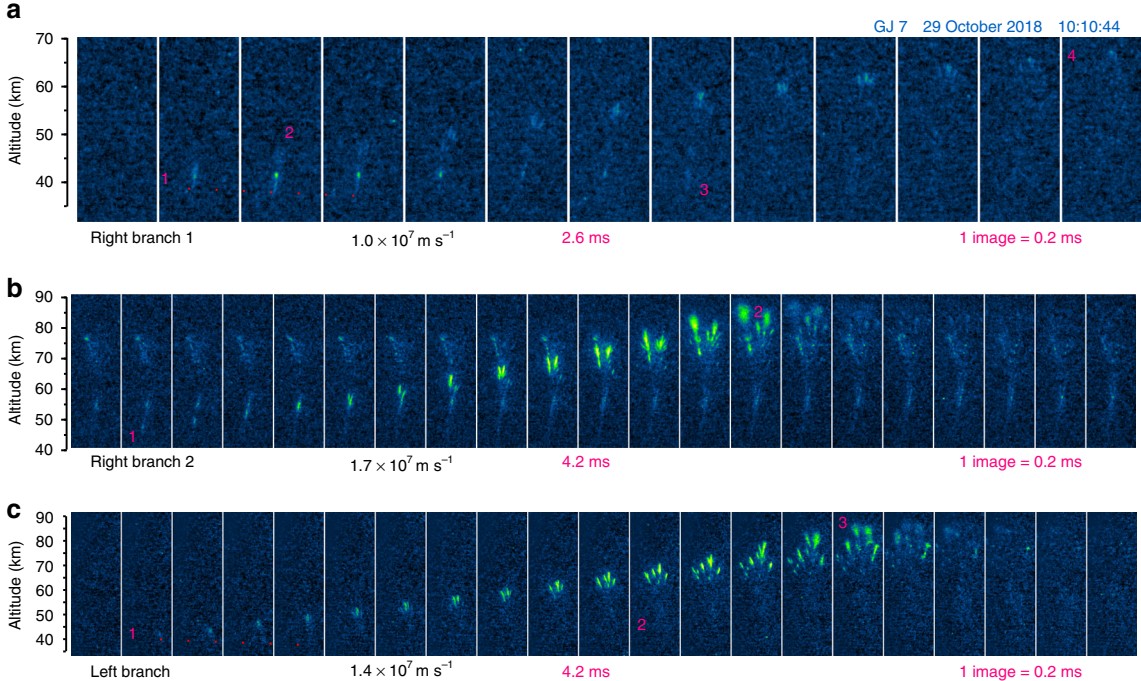

**Fig. 3** Development of the three branches of gigantic jet 7 recorded by the high speed camera. Image sequences recorded at 0.2 ms intervals, showing the upward speed of the development, and their brief downward extending parts indicated by a dotted line. **a** the first right-hand branch, with a clear bidirectional onset, **b** the second right-hand branch, overlapping visually with remnants of the first, **c** the third branch, occurring to the left of the others, with a subtle downward component

previous segment was initiated (32.6 km). Here, the upward negative streamer heads leave a rather strong, persistent glow in their wake. A similar evolution is observed in the left-hand branch of GJ 7 (Fig. 3c). The second branch development of GJ 7 (Fig. 3b) overlaps with the afterglow of the first, and its initiation altitude was 45 km, a few km higher. Remarkably, Fig. 5b (and Supplementary Fig. 6) shows that the third branch development, to the left of the previous branches, appears to have suddenly lowered the luminosity of lasting beads or segments in the right-hand branch.

The typical duration of the final jump is about 2 ms, with measured average speeds of $1.8 \times 10^7$ m s$^{-1}$ (GJ 3), $2.3 \times 10^7$ m s$^{-1}$ (GJ 4), 1.0, 1.7, and $1.4 \times 10^7$ m s$^{-1}$ (separate branches of GJ 7), and $2.5 \times 10^7$ m s$^{-1}$ (GJ 12). Figure 2 displays the highest speed between subsequent 1.1 ms frames for GJ 3 and 4. It can be noted by the slope that the speed in GJ 7 and 12 is almost doubled above 70 km, with a maximum of $7.5 \times 10^7$ m s$^{-1}$ (GJ 12, 0.2 ms before reaching FDJ) while the streamers widen considerably. Upon the faster branches reaching the ionosphere, in GJ 4 (at mark 3) and GJ 12 (at mark 3-4 in Fig. 2d) the slower branches bend and connect under straight angles to these fast branches. This behavior is identical to laboratory streamers bridging a gap[31].

**Trailing Jet stage**. After reaching the ionosphere, none of the events show an immediate increase in brightness along the jet to indicate a return stroke as in cloud-to-ground lightning. The upper jet just decays slowly, while the middle part of the jet starts to form a brighter section after a few milliseconds. In GJ 3 (Fig. 2a), the fork stem (4) at 50 km briefly increases in luminosity upon GJ completion (5). The lower fork (3) does not get this boost in brightness. From (6) onwards, the lower half of the GJ increases slightly in brightness, while the stem of the upper part of the jet decays to a minimum at (7). From (7) onwards, the top of the TJ becomes brighter while rising. GJ 4 and 12 are similar,

but the trailing jet onset is faster, shortening the dark period after the FDJ. The dark intermezzo without a clear development lasts roughly 11 ms (GJ 3), 6 ms (GJ 4), and 4 ms (GJ 12) which is best visualized by Figs. 4 and 5.

Most of the luminosity of the lower jet channel is produced up to +60 ms relative to the FDJ, the maximum being around the +20 ms mark. This is around the time the bright top (transition zone[12,17]) of the TJ slows down from a relatively fast $5–9 \times 10^5$ m s$^{-1}$, to a slow pace of $1.6–2.3 \times 10^4$ m s$^{-1}$. This change in velocity can be appreciated by the curved section in the time-altitude Figs. 4 and 5 and occurs in all cases around 50 km altitude. The fast TJ top rising stage (when luminous) lasts about 8, 10, and 12 ms in GJs 3, 4, and 12 respectively. In GJ 7 it rises slower but longer (~25 ms).

The TJ tops out at 55 km (GJ3), 58 km (GJ4), and 62 km (GJ 12). In GJ 7, each of the three final jumps (FDJ) is matched by a corresponding TJ feature as indicated in Fig. 5b, topping out at 64, 62, and 57 km respectively. It can be noted how the delay between TJ brightness and the jet it follows up is ~20 ms. It seems not to matter whether the FDJ was incomplete (first branch of GJ 7), for a TJ transition region to form. Similar to the events observed from Réunion Island[17], two surges in TJ brightness are observed during GJ 4 (Fig. 4b), likely in response to in-cloud lightning processes that change the potential of the channels. The brightness moves upward as a wave and the height of the TJ top increases from 55 km to 58 km.

Finally, by backward extrapolation of the TJ transition zone in the time-altitude plots it can be noted how the beginning of the transition zone of the TJ appears to originate from the altitude of the initiation of the final jump, although the transition zone starts to glow brightly usually several kilometers higher. The width of the transition region and its number of beads corresponds to the branched structure of the FDJ earlier at that altitude, as also clearly seen in the events of Soula et al[17].

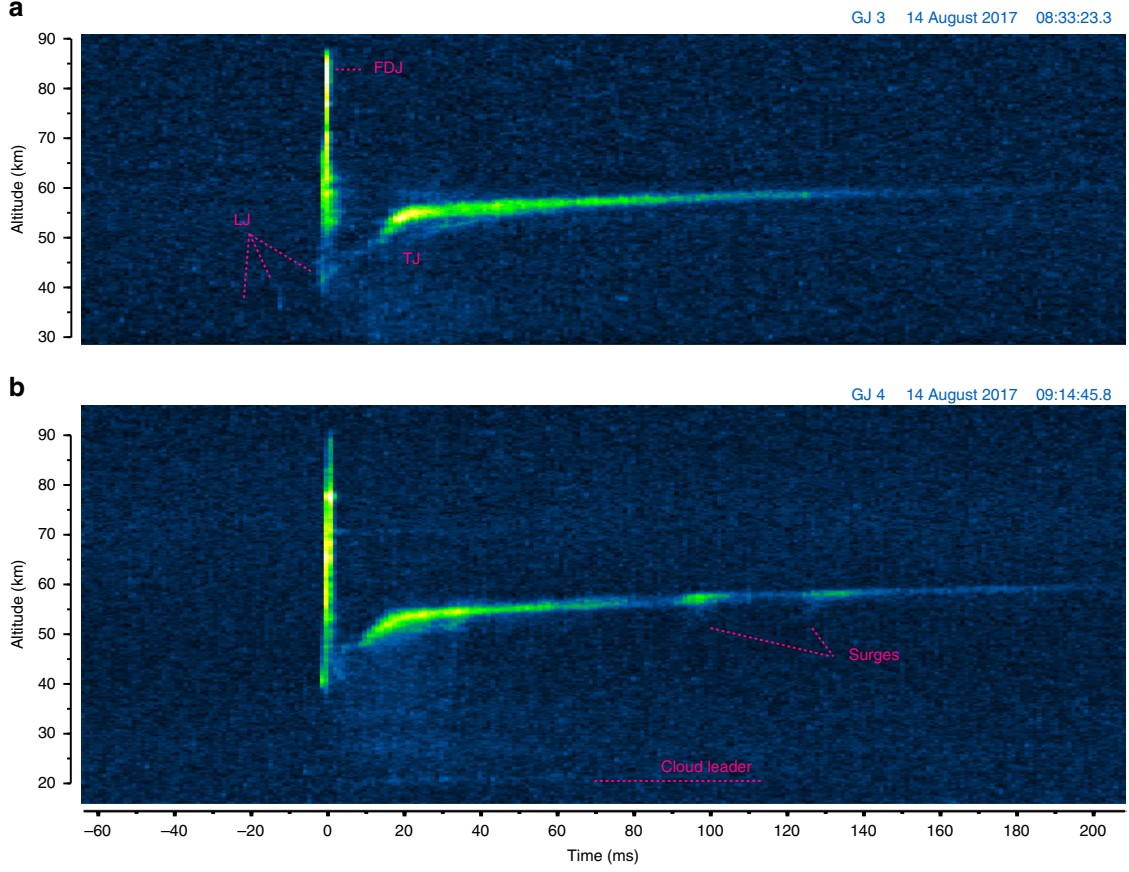

**Fig. 4** The evolution of brightness with altitude in gigantic jets 3 and 4. Time-altitude-luminosity graphs derived from high speed video images of the jet events with a temporal resolution of 1.11 ms. The graphs are scaled such that equal slopes correspond to the same speed across the events. **a** GJ 3 with the Leading, Fully Developed, and Trailing Jet stages indicated. **b** GJ 4. Note that the section below 30 km in GJ 3 was not within the view of the high speed camera

**ELF radio signature and Global Lightning Mapper.** We now compare the evolution of luminosity of GJ 12 in the altitude ranges marked by ABCDE in Fig. 5a to magnetic field signals recorded by stations in Cape Verde and Duke University (see Methods). We include the luminosity recorded at 2 ms intervals in the 777.4 nm wavelength by the Geostationary Lightning Mapper (GLM)[38], emitted by atomic oxygen emissions in hot lightning channels. Figure 6 shows the evolution of the brightest GLM pixel. Similar to Boggs et al.[34], we find that two adjacent pixels (about 14 × 7 km), the brightest one corresponding to the GJ azimuth, are consistently brighter than surrounding ones.

The GLM curve matches very well the evolution of cloud luminosity detected by the high speed camera (E). The vertical dashed line marks the time the FDJ is completed. The cloud luminosity picks up just after this time. The optical energy peaks at 32 fJ. Due to automatic thresholding in GLM[34], brightness of the cloud became more reduced compared to the ground-observed luminosity from about 30 ms after the FDJ, displaying a stepwise decrease towards zero.

The luminosity in the lower jet section with the step (D) picks up immediately after the FDJ time, while the transition zone (B) and top (A) were still decaying. Radio signatures of Cape Verde and Duke both reveal a broad maximum lasting about 30 ms (full width at half maximum) corresponding well with the shape of the cloud flash curve. A very slight increase in the magnetic field waveforms appears during the LJ stage, followed by a sharp peak, especially in the Duke waveform, centered on the FDJ time. Because of its lower bandwidth, the waveform from the Cape

Verde receiver shows a more subdued peak, lagging slightly relative to Duke.

The precise time-resolved, GPS-time referenced gigantic jet recording allows the conclusion that the 2 ms rise time to the peak in the magnetic field is produced during the 2 ms duration of the final jump itself, and the signal drops for 3 ms after the streamers reach the ionosphere, which corresponds to a continued glow in the upper jet (Fig. 2c, d) until the onset of the TJ. The following 30 ms broad peak in the signal corresponds to a current between cloud and middle parts of the jet, which starts at the time of the FDJ. The increase of this signal coincides with the fast rising stage of the transition zone of the TJ and its maximum brightness.

The charge moment change up to the final jump is 25 C km. The final jump and decay of the FDJ contribute 40 C km combined. The current when the LJ reaches 25–30 km is estimated at 100 A (2 kA km over 20 km channel length). During the final jump the current increases to 350 A. In all, 0.5 C of charge is displaced upward during this jump along the entire jet, using the average charge altitudes at start and end of the jump assuming a uniform distribution along the channel. The highest current moment during the TJ is 37.5 kA km. Assuming most of the current to flow between the negative cloud charge at ~8 km[39] and TJ top at 50–55 km at that time, this stage produced about 850 A. The current density at 30 km altitude where the channel diameter was 550 m is $3.5 \times 10^{-3}$ A m$^{-2}$. The re-activation of beads and new patches in the upper jet could be a result of the enhancement of electric fields in the mesosphere as result of large

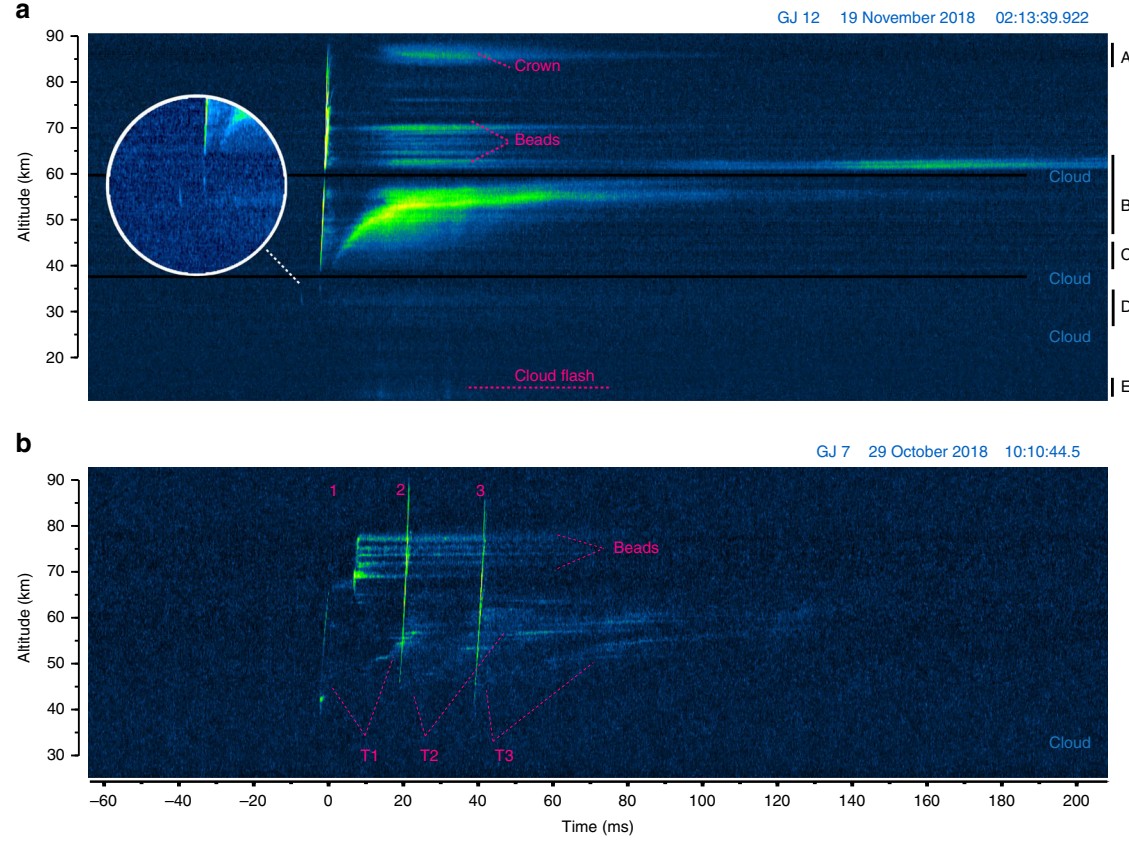

**Fig. 5** The evolution of brightness with altitude in gigantic jets 7 and 12. Time-altitude-luminosity graphs at 0.2 ms temporal resolution for **a** GJ 12 and **b** GJ 7. The letters A–E indicate the altitude ranges used hereafter to track the evolution of luminosity

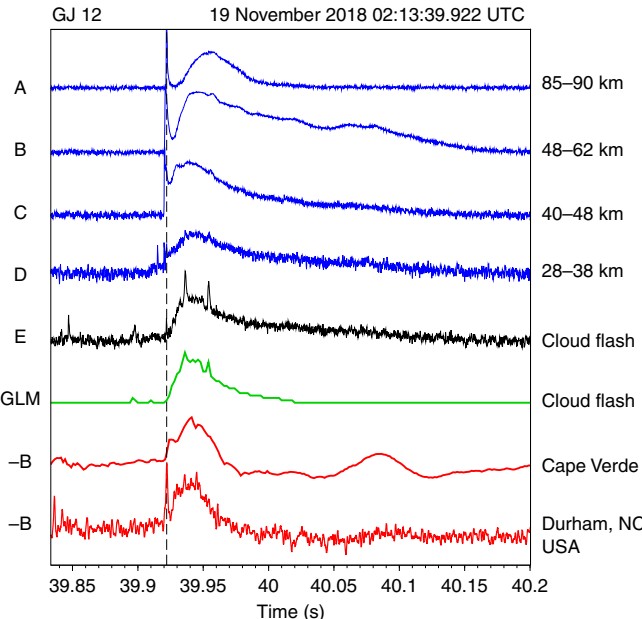

**Fig. 6** The brightness evolution of sections of gigantic jet 12 compared to magnetic field recordings and the parent cloud flash. Altitude sections of the jet are represented by A–D. Brightness is scaled to avoid overlap between curves and enhance weaker features in D and E. Cloud brightness (E) has been recorded by the high-speed camera and by the Geostationary Lightning Mapper (GLM) on the GOES-16 satellite. The bottom rows show the Extremely Low Frequency magnetic field waveforms from Cape Verde (north-south component) and Duke University (east-west component) orthogonal to the event direction

vertical charge transfers, like in the case of sprites[40,41]. The total charge moment change by the end of the TJ stage is 2300 C km.

## Discussion

The first high-speed camera recordings reveal more than a dozen new features in the evolution of gigantic jets, summarized as follows: first, a stepping process occurs during the late leading jet stage, here with step sizes of 2–5 kilometers, 5–10 ms intervals, 0.5 ms step duration, and $>9 \times 10^6$ m s$^{-1}$ step speed. The final jump develops from a bidirectional onset at ~35–42 km altitude and propagates continuously toward the ionosphere, starting the fully developed jet stage. The upward negative streamers propagate at mean speeds of $1-4 \times 10^7$ m s$^{-1}$, accelerating above 70 km and leaving a glow in their wake. No return stroke is detected upon connection. Late streamers bend and connect to early ones. The transition zone of the trailing jet can be traced back to the FDJ onset, but only becomes bright after a delay of 5–11 ms. A high resolution camera made it possible to identify a narrow, bright lightning channel at the root of the lower jet during the TJ stage in two cases, and its rebrightening of 500 ms duration in one of these cases. One of the jets had an appearance similar to carrot sprites in the upper section. Its beads rebrightened and new patches appeared at the top during the trailing jet stage. The optical evolution of this jet and its cloud flash are now matched precisely with features in the vertical charge transfer. In another event three separate branches developed. The slowest of these stalled in the mesosphere, and three corresponding transition zones formed. Luminosity of persistent beads in the upper right-hand branch suddenly dropped during the development of the new branch to the left.

In the gigantic jet events of this study, two events exhibited one or more steps prior to the final jump, while GJ 4 did not reveal any steps despite good lower jet visibility later during the event. Stepping is a well-known characteristic of negative leaders in lightning flashes, and our observation could be regarded as confirmation for the hypothesis of the lightning leader reaching ~40 km altitude in gigantic jets, after which its streamer zone can make the jump to the ionosphere[8,18]. In such case, the observed segment would be a pulse of current (stroke) after a space leader in front of the leader tip connects to the leader channel (e.g. refs. [42,43]), as this emits the most light during a leader step. However, the 3 frames (Fig. 2c) clearly show a much longer process, less bright than the ensuing streamers of the final jump, a few kilometers higher. The low visibility of much of the LJ in the high-speed images could indicate that optical emissions were produced mainly in the blue/ultraviolet part of the spectrum which is filtered out strongly by the atmosphere over the distance of ~350 km. Emissions from visible and near-infrared wavelengths appear to be lacking, while those are the ones normally abundant in leaders in the lower troposphere (e.g.[44]). Similarly, photometers in space[15,45] demonstrated no clear 777.4 nm O (I) emission in gigantic jets, which is a common spectral line in lightning. Likewise, 777.4 nm emissions during the LJ stage in GJ 12 were too weak to become detectable by the GLM, if present at all. The GLM brightness shows virtually the same evolution as the cloud flash brightness in the high speed camera, as expected based on its filter design.

Stepwise propagation is not unique to negative leaders, however. It has been described earlier in embers[46], a form of upward secondary discharges under sprites[47,48] some of which resemble jets[49] of negative polarity. We obtained a new high speed, short range recording of this phenomenon in Colombia, described in the Supplementary Discussion. Similar to embers, and to negative laboratory discharges[26,50–52] the final jump in two gigantic jet cases (GJ 7 and 12) is clearly observed to start bidirectionally, with negative streamers growing upward and positive downward (from a bright node in GJ 7 – which may be a space stem). The downward streamer could not be traced all the way down. It was retracing the previous step in GJ 12. In neither case this was followed by any visible step-like pulse which would occur if connecting a space leader to a main leader channel below. By time-altitude plots, the TJ top can be traced back to the FDJ onset. Its brightening occurs at the same time as the maximum cloud brightness, and coincides with magnetic field recordings indicating a strong continuing current. During the TJ, these transition zone beads could be likened to the space stems in long negative discharges shown by Les Renardières Group[50], and the weaker TJ channel to its positive streamer zone reaching backwards to the leader tip. Like the top of the TJ, the laboratory space stems also drifted slowly between steps.

The high-resolution camera recorded a bright lightning-like channel at the bottom of the jet in GJs 3 and 4 (Fig. 1), which remained narrow and flashed brightly after the end of the TJ. It ultimately extended up to 22–23 km altitude. Considering this aspect and the features of the one resolved step, it is likely that the LJ is a streamer corona extending from this cloud leader which initially was not yet visible above the anvil cloud (see also Supplementary Fig. 5). This scenario is also plausible among the modeling scenarios of Da Silva & Pasko[9], described by their fig. 20b: the streamer corona extends kilometers higher than the leader tip, and accelerates into a final jump from about 40 km altitude, while the simulated leader tip at that moment reached 28 km. With a higher leader potential this situation could occur for leaders at lower altitude. A final leader altitude of 20–22 km and a speed of 4 to $6 \times 10^4$ m s$^{-1}$ of this bright section is consistent with their (fig. 15) 1 A scenario[9]. This current applies for a

0.3-mm sea-level equivalent $1/e$ leader radius, scaling with atmospheric density, but the authors argued that the diameter may well be larger in the real world, and emphasized that current density is the physical factor that determines heating and propagation. So, with the same current density, a 3–4 times wider leader channel would carry a current closer to 100 A.

The velocity of the final jump ranged between 1 and $4 \times 10^7$ m s$^{-1}$. This fits exactly into the range of values found[15] using the photometer array on the Imager of Sprites and Upper Atmospheric Lightning (ISUAL) instrument on FORMOSAT-2. The lowest speed final jump in our case stopped ~20 km short of reaching the ionosphere, but triggered a new upward development ~7 ms later to reach 10 km higher. This part bears some similarity to the ember discharge (Supplementary Figs. 10 and 11). The low speed coincided with weaker branching and likely signifies it was propagating under a weaker electric field than the others.

Upon reaching the ionosphere, none of the events show a return stroke moving downward from 40 to 50 km altitude as detected from space[15,45]. The likely reason is that it occurred in UV wavelengths (337 and 317.6 nm) to which our intensifier is not sensitive. The downward positive streamer in GJ 7 and GJ 12 cannot not be traced below 35 km altitude, but starts before the FDJ time, not ~1 ms after. A dark period of 4 – 11 ms follows the FDJ with only slight luminosity in the middle jet, after which the TJ forms with a bright top. It lasts too short to be detectable in normal speed video. It was postulated that the ionosphere's lower boundary is brought downwards during the FDJ to the middle part of the jet, which would explain the TJ as a continuous arc of hot leader channels between capacitor plates[15]. However, the case of GJ 12 clearly shows that an electric field is present above the TJ, as beads and new patches appear. This could indicate that such lowered boundary either does not exist, or dissolves very rapidly, before most of the TJ. This is also reflected in the fact that a completed path to the ionosphere is not necessary for the TJ feature to develop.

The cause of the sharp peak in the magnetic field signal has now been unambiguously defined as the fast ascending streamers during the final jump (lasting 2 ms) and decay of the FDJ (3 ms), thanks to the unprecedented precise timing of the images. The upward motion of the TJ top slowed down from $9 \times 10^5$ m s$^{-1}$ to $2 \times 10^4$ m s$^{-1}$ after the maximum of the broad peak in the current moment was reached, which corresponded well with the cloud luminosity. The maximum current density at 30 km altitude during the TJ was $3.5 \times 10^{-3}$ A m$^{-2}$. A leader at that altitude should reach ~$2 \times 10^3$ A m$^{-2}$[8,9], but our value is 6 orders of magnitude below that. During this time, however, the jet channel was brighter than during the LJ stage, and as such it is unlikely that the current density would have been higher during the LJ even though the channel was still narrow.

In the Supplementary Discussion we show how existing electrostatic models for gigantic jets[5,6,8,9] can be combined with stepping in the negative streamer zone[53]. The combined model offers an improved explanation for the morphological differences among negative gigantic jets, including the variations of final jump altitude, as a function of leader tip altitude and potential. The observed features in the evolution can help guide more advanced models to complete our understanding of these electrical phenomena and their impact in the Earth's atmosphere.

In conclusion, LJ steps (and absence thereof) and the bidirectional onset of the FDJ have properties (a long step duration of 0.5 ms, no optical pulses upon connection) which are difficult to unify with the recently arisen paradigm that the LJ is a lightning leader to about 40 km altitude. These features and the beads in the TJ top are plausible manifestations of bipolar space stems known

from negative laboratory discharges[26,48–50,53] without converting fully into hot leaders.

## Methods

**Optical instrumentation**. An observation campaign was conducted near Santa Marta, Colombia (11°8'N, 74°13'W) from 29 July 2017 to 23 August 2017. The video system consisted of a Vision Research Phantom Miro 3 fitted with a Gen III image intensifier sensitive to light with wavelengths from 400 to 900 nm, with P-43 phosphor (1 ms decay time), a Nikon 85 mm F1.4 G lens to focus its image on the sensor, and a Nikon 28 mm F2.8 input lens for approximately 28 by 17 degrees wide view. The Miro was set to record 900 images per second. Its memory was segmented into 3 parts storing video data from a circular recording buffer upon trigger. The relay lens was stopped down to f/2.8 and the input lens to f/3.5 to ensure sharpness across the image, at cost of lower signal to noise ratio. At the image size of 800 by 600 pixels used, the spatial resolution of this setup was 130 arcsec pixel$^{-1}$. The camera was aimed manually over distant storms and triggered by an audio signal provided by UFOCapture video event detection software (by SonotaCo) running on a laptop processing the video stream of a bore-sighted Point Grey Grasshopper USB 3.0 camera with a monochrome Sony IMX174 global shutter CMOS sensor, fitted with a Navitar 25 mm F0.95 lens with almost the same viewing angle (25.3 by 15.8 degrees) as the intensified Miro. This camera ran at 1920 by 1200 pixels at 20 frames per second (GJ 2 and GJ 3) or in 2 × 2 binning mode at 960 by 600 pixels at 60 frames per second (GJ 4). GJ 3 was resolved by this camera at a resolution of 47.4 arcsec pixel$^{-1}$, translating to about 80 meters pixel$^{-1}$ at the jet location.

In the 2018 campaign (GJ 5–12), the detection camera was used in binning mode at 50 frames per second and the high speed camera was the Vision Research Phantom V7.3 running at 5000 images per second, fitted with a Gen III image intensifier (400–900 nm with most sensitivity between 500 and 900 nm) with fast P-24 phosphor (1 μs decay time). A Nikon 50 mm f/1.8 was the input lens. The angle of view was 21 by 13 degrees (without image corners), resolving 95 arcsec pixel$^{-1}$. IRIG-B (Global Positioning System) timing with an accuracy of a microsecond was available for GJ 12, the other events were timed by Network Time Protocol with a typical accuracy of 0.1 s.

**Image processing**. Image sequences have been processed in (Fiji) ImageJ software[54]. Uneven backgrounds have been subtracted by creating a median image of frames around the event (Z project), subtracting 10, and subtracting this image from the stack. Temporal background oscillations from city lights were removed by the "Subtract background" tool which subtracts a regional mean value from each pixel per frame. Hot pixels common in the intensified images have been removed by the "remove outliers" function with appropriate threshold maintaining GJ features as much as possible. Time-altitude-brightness figures (Figs. 4 and 5) have been created using the kymograph analysis tool, based on a segmented line drawn along the vertical path of the jet filaments. Background glow was subtracted first. Line width was 1 pixel, we shifted the line sideways in steps. We then took the maximum of resulting kymographs to ensure including optimally the GJ features. Horizontal banding from the camera or kymograph result was reduced by a Fourier band-pass filter or a Fourier/wavelet/moving average plugin.

Stacked images from the Grasshopper were automatically matched to the star field via *nova.astrometry.net*, resulting in precise elevation readouts in Sky Charts/ Cartes du Ciel v4.1. The altitude calculation has been described in the supporting information of van der Velde and Montanyà[55] and uses the local Earth radius in east-west and north-south directions based on WGS84. Feature altitudes of GJ 3 and 4 are precise to about 0.5 km altitude thanks to lightning detections unambiguously associated with the events, obtained from Colombia's *Keraunos* network. These detections were clusters of <1 km across, located directly under the gigantic jet azimuth. For GJ 7 the GLM and cloud top helped narrow down the distance range, with altitude error margins of 2 km. GJ 12 observed from Cartagena was triangulated together with another star-referenced image from Santa Marta. Its altitude error margin is ~1 km. Possible tilt angles of the events along the viewing direction may result in larger error margins.

**Electromagnetic field measurements**. A pair of induction coil magnetic field sensors at Duke University (35.971°N, 79.094°W) and a similar receiver (LEMI-419) at the island of Sal of Cape Verde (16.73°N, 22.93°W), with a flat frequency response between 2 Hz and 25 kHz (Duke) and <0.01 to 300 Hz (Cape Verde) recorded the vector horizontal magnetic field produced by the gigantic jets. The current moment waveform, and thus time-integrated charge moment, were extracted from the azimuthal component of this magnetic field from Duke with the approach used by Cummer et al.[14]. Supplementary Figs. 7 and 8 show the fit to the data and the resulting current and charge moments. All waveforms of Fig. 6 have been corrected for propagation times between observer (speed of light, $c$) or radio receiver (0.9 $c$) and the jet event. In case of GLM this correction is already included with the data. The error margins on the estimated current in the jet are mainly those associated with the raw measurement (~10%) and the model-based inversion to obtain the current moment (~25%). The uncertainty about the altitude of the negative charge source adds 5–10%.

## Data availability

The original video data and related files of this study have been deposited in a permanent scientific data repository: https://doi.org/10.5281/zenodo.3353728[56].

## Code availability

The code for the electrostatic model described in the Supplementary Discussion is available from the corresponding author.

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

## Acknowledgements

This work was supported by research grants from the Spanish Ministry of Economy and the European Regional Development Fund (FEDER): ESP2013-48032-C5-3-R, ESP2015-69909-C5-5-R, and ESP2017-86263-C4-2-R. S.A.C. contribution was supported by the US National Science Foundation Dynamic and Physical Meteorology program through grant AGS-1565606. O.A.V. acknowledges discussion with S. Nijdam, U. Ebert, A. Luque, F. Gordillo, T. Neubert, and N. Liu. We gratefully acknowledge the hosting of our high-speed camera system at the Meteorological Department of Curaçao (A. Martis and H. Pieter) during the years prior to the Colombia campaigns.

## Author contributions

O.A.V. conducted the observations, processed images, determined feature altitudes, drafted and revised the manuscript text and figures. J.M. implemented the fast camera system, installed and processed data from the Cape Verde receiver and managed the project. O.A.V. and J.M. programmed the electrostatic model and developed the discussion. J.A.L. assisted with the campaign and observations in Colombia, retrieved lightning detection data for the events and investigated leader potential using a model. S.A.C. retrieved and processed Duke receiver data. All authors discussed and support the conclusions.

## Additional information

**Competing interests:** The authors declare no competing interests.

