## [Peer Review File · Nature Communications]

Reviewers' comments:

Reviewer #1 (Remarks to the Author):

Review of "Gigantic Jets reveal a stepping process in streamers in the stratosphere" by van der Velde, Montanyà, and López

This manuscript reports an observational study of gigantic jets by using a high-speed camera in conjunction with a sensitive but slower camera. Four GJs were observed on two nights in 2017, and two of them are discussed in detail in the manuscript. It is concluded that the leaders of the GJs reach only 21 km altitude, and above that altitude the GJ discharge develops in a stepwise manner similar to the "pilot system" propagation mode in negative streamers.

As far as I know, high-speed images of gigantic jets have never been reported. From this point of view, the results reported in the manuscript might deserve to be published in a high-profile journal such as Nature Communications. However, I do not think that the conclusions drawn in the manuscript are fully substantiated by the results presented, which is explained in detail below.

General Comments

1. After carefully reading the manuscript, it appears that the conclusion that the leader discharge only reaches 21 km altitude is drawn based only on the images showing that the GJ channel at this altitude is thinner and brighter. Given that the events occurred at a distance about 350 km from the observation site, I do not think this argument is convincing enough. It is entirely possible that the brightness of a single leader is under the detection limit from this distance in the humid air condition of tropics. The leader tip stops propagating upward after exiting the cloud top is also inconsistent with previous GJ observations [e.g., Pasko et al., 2002; Su et al., 2003; Soula et al., 2011; Liu et al., 2015]. In addition, there may be other reasons why the brightness of the channel section just above the cloud lasts longer. For example, the channel just below and above that section has more branches.

2. As suggested by Raizer et al. [2006, 2007], in order for the discharge channel to maintain its conductivity on the GJ timescale of a few milliseconds to tens of milliseconds, the discharge channel needs to have a high temperature. The cold streamer channel cannot sustain its conductivity too long to continue the upward development of the GJ. Detailed theoretical and modeling studies [Riousset et al. 2010; da Silva and Pasko, 2012, 2013] also indicate that the streamer-to-leader transition can and should occur beyond 21 km altitude for the timescale of GJs. The authors need to address if there are only streamers above 21 km altitude, why the GJ can last that long.

3. In the simulations presented in the manuscript, the potential of the leader tip is more than 150 MV. At such a high potential, it is hard to understand why the leader does not continue propagating upward but just stays there. For the stepping model, if the field above the leader tip in the end is maintained at the positive streamer propagation threshold value, the question is again why the discharge plasma does not quickly decay due to attachment and recombination.

Detailed Comments

L16-19: See my general comments. The stepping feature observed may be just associated with the propagation of negative stepped leaders.

L19-20: This is a bit speculative. More detailed comments are given below.

L42-44: Jets can reach this altitude range without developing into GJs. This statement is not correct.

L61-62: A 0.25 km uncertainty is very small. The GJ channel above the cloud may not be at the same horizontal location as the associated lightning, and even the detection of the associated

lightning may have a large location uncertainty.

[SEP]L63: List an altitude scale on Figure 1.

[SEP]L72-77: Light is severely attenuated by the high humidity and haze conditions in tropics. The events occurred very far away from the observation site, and it is possible that a stepped leader at such a large distance is below the detection limit of the camera. The long-lasting pixel may indicate that there are more leader branches below and above this cloud exit point. I am not convinced that the leader only reaches about 21 km altitude.

L77-80: How do you know the tropopause height is 16 km and the temperature was -77 C? Was it from a balloon? If so, provide the balloon sounding, where it was launched from, and the time of the launch. Also, the coldest satellite cloud top temperatures were about -85 C for the June 30 event, and about -78 C for the August 14 events (see attached images). Thus, the cloud top was at/above the tropopause for all events.

L84: Give what UTC times correspond to the panels in Figure 2a, b. You have the 0 mark for milliseconds on the x axis, but what UTC time does this correspond to for both GJs? Also, put both the detailed evolution of the GJs (the high-speed images) in the Supplementary file.

L89-90: Consider to add a time series of the whole GJs from start to finish for the high resolution (low speed) camera, especially since the fast camera doesn't catch much luminous activity before the FDJ stage.

[SEP]L92-94: I don't understand the statement. How is the average speed limited by the long exposure time of the slow camera? Is it based on the assumptions that the discharge is stepping? Do you mean the average stepping speed? This is a typical average speed for leaders, but 0.4×10^5 m/s is too small for streamers.

[SEP]L98-101: First, those segments could be leader channels; Second, because the discharge is barely detectable, this speed may not correspond to the upward extension of the channel but fast luminosity variation of existing channels. Third, it is not "almost two orders of magnitude" but slightly more than an order of magnitude.

L141-144: High-speed images of downward stepped leaders of CGs show the luminosity of a negative stepped leader is non-uniform along the channel, and also varies over time. The text here is inaccurate.

[SEP]L144-146: Well, light from the channel at a lower altitude may be scattered and absorbed more severely because of the slanted path passing through a longer distance in denser air.

[SEP]L146-156: First, the duration of the plasma created by streamers should be checked to see if a pilot node can exist that long. Second, as mentioned above, the higher observed speed may not correspond to the upward propagation of a discharge channel.

[SEP]Stepping Model: This model may be overly simplified to describe GJs. It is basically a static model. The authors need to first validate the model by comparing some common parameters that are reported by previous modeling studies of GJs, leaders, or streamers. Second, the timescales of the process involved should be discussed and should match the timescale and speed of GJs.

L168: For negative leaders, U_0 has a negative value. Do you mean $|U_0|$? This should be clearly indicated.

[SEP]L179: For such a large potential, why does the leader tip not continue propagating upward? This needs to be explained.

[SEP]L191: 'but not below (for reasons explained before)' --- what are these reasons? Because of attenuated light? Please make this clearer here.

[SEP]L194-195: These potentials are very large and may be unphysical. Additional explanation is required to show that those potentials are realistic for leaders.

[SEP]L218-222: See my general comments above. To conclude that the gigantic jets are streamers, those three comments need to be clearly addressed. The last sentence of this paragraph also somewhat contradicts with this conclusion because the leader during LJ stage is not bright enough to be seen and it is entirely possible that the leader reached higher altitude. To definitively say where the ascending negative leader stops and the streamers dominate requires more evidence than a long lasting pixel above the cloud during the FDJ stage.

L227-223: Production of X-rays and Gamma-rays by leaders or streamers doesn't necessarily lead

to TGFs. This is too speculative.

7-30-2017

06:30 UTC CH 13 infrared

Kelvins

Range and Bearing

Distance: 362.89 km

Azimuth: 199.9

Back azimuth: 19.7

8-14-2017

08:30 UTC CH 13 infrared

Kelvins

192.2

301.2

Range and Bearing

Distance: 354.51 km

Azimuth: 195.7

Back azimuth: 15.6

CMI - Color-Shaded Plan View
Range and Bearing

Time = 2018-02-25 19:39:43Z
2 of 2

Reviewer #2 (Remarks to the Author):

Review of the manuscript titled "Gigantic Jets reveal a stepping process in streamers in the stratosphere" submitted for publication in Nature Communications by van der Velde et al.

The reviewed manuscript presents for the first time gigantic jet (GJ) observations with 1 ms temporal resolution. The authors use the combined information from two cameras, one with high temporal, and another with high spatial resolution, to infer the propagation mode of the discharges in the stratosphere. The key conclusion of the paper is that GJs are mostly streamers, and a lightning leader channel only reaches up to ~20 km altitude.

Although the observations are very interesting, the authors make a lot of conclusions in the paper that are not supported by the images shown. The key issue here is the argumentation that GJs are mostly streamers just because they don't look like lightning leaders in the troposphere. This is a tricky statement to be made. Other factors need to be considered, such as the electrodynamics involved and the plasma composition. These factors can only be probed with electromagnetic field measurements and with spectroscopy, which are not available in the present investigation. Moreover, the numerical model presented does not seem to be physically correct. The issue may only be the lack of proper description. But with the (little) information given in the paper, this reviewer can only conclude that the model is wrong.

My overall impression is that the interpretation is not supported by the available experimental data and by the simulation performed. There are a lot of conclusions that cannot be seen from the four figures in the paper. For all these reasons, I am inclined to reject this paper. But at this time I will recommend the paper to be returned to authors for major revision. The paper may be suitable for publication if it is reframed as "the first high speed observations of GJs" and more careful conclusions are made about the discharge physics. Based on previous work of the same authors, I was also expecting to see a more illustrative analysis of the meteorological scenario involved in the production of these GJs.

Below you may find additional items to be considered:

1. Line 1: Title. "Gigantic Jets reveal a stepping process in streamers in the stratosphere". The terminology streamer stepping means something else in laboratory discharges. You can see streamer stepping in a single streamer channel, where subsequent discharges reuse the same channel and propagate further than the 1st one. What you see in the images is not streamer stepping, but the complicated dynamics in a streamer zone of a lightning leader. Negative lightning leaders likely step, i.e., pause for some time before propagating again, as a result of this complicated dynamics.
2. Lines 15-16: "... At a temporal resolution of 1 ms ..." The authors should clarify here that the key conclusions of the paper do not come from the camera with 1 ms temporal resolution. The authors state that the initial propagation is in the streamer and not leader mode. But that conclusion comes from the lower temporal resolution camera. The high-speed camera doesn't even see the initial propagation.
3. Lines 41-44: "The decrease of air density ... jump to the ionosphere." Please add reference to this statement.
4. Figure 2: I'm assuming that the images shown in panel (2) are from the fast camera, which has 1.1 ms temporal resolution. Where are the missing frames? They need to be shown here to justify the conclusions. Is there anything present in those frames?
5. Figure 2: The dotted arrows in panel (2), I believe, indicate the propagation of the discharge. Is the reader expected to see the bidirectional propagation described in the text from this figure? That is not possible. There is no clear bidirectional propagation in this figure.
6. Figure 2: Features that are only slightly above the noise threshold of the camera need to be taken with a grain of salt. Please do not base the main conclusions of the paper on such features.

7. Line 127: "After this darker period, beadlike structures ..." This statement is very interesting. However, it is not supported by Figure 2. The formation and propagation of the described bead-like structures needs to be shown in the imagery.

8. Lines 139-156: The entire Interpretation section needs to be revised. The key idea that an inferred stepwise propagation is associated with the formation of a pilot system reinforces the idea that a lightning leader channel is involved and not the other way around. I would not expect leaders in the stratosphere to look like the ones in the troposphere.

9. Lines 158-211: The entire Stepping Model section needs to be revised. It is very difficult to connect Figures 3 and 4 with the Gallimberti [2002] model which the authors say that they are using (which anyways is an obsolete framework). But if you look at Figure 14 (left) of Gallimberti [2002] you can see that after the emission of the streamer corona flash from the leader tip, the potential in the streamer zone rises, so that a lower electric field E_{cr} exists in the streamer zone. After relaxation and emission of a bidirectional streamer corona, the potential rises even further. Nothing of this sort is seen in Figures 3 and 4. The authors need to describe their modeling in more detail, maybe in the Supporting Information manuscript. Perhaps showing profiles at different time instants illustrating the different stages of the bidirectional corona development during one of the steps.

10. Lines 158-211: In the early models, the need for having a leader propagating up to some altitude in the stratosphere appeared in order to allow for GJs to be formed from realistic voltage drops. In the proposed model, the authors find that 300 MV is required when adopting realistic values of E_{cr} . This should be seen as a red flag for the proposed simulation methodology. What is the maximum realistic value for the potential difference between charge layers in an entire thunderstorm?

11. The two montages included as supporting information (SI) need to be included/described in the SI manuscript.

Responses to reviewers (in bold)

General overview of revisions:

- 1) The introduction includes an overview of leader/streamer features in past publications and discusses the use of speed to distinguish between these.
- 2) The gigantic jet figures have been completely redone. They now show sequences with features labeled with numbers described in the text, with a better color scale.
- 3) A figure with satellite images, lightning detections and the direction of the GJs was added.
- 4) A figure with a secondary jet (ember) event was added, recorded during the 2017 Colombia campaign. This event shows details of stepwise propagation in a streamer discharge.
- 5) The conceptual model and its figure has been described better. Its discussion was simplified. A graph showing the electric field at the streamer tips was added, which helps explain its acceleration.
- 6) Discussion is updated to address the leader/streamer identification and the feasibility of high leader potential.
- 7) Materials for reviewers include a PDF with leaders/streamers indicated in past publications and recent online photographs, as well as the code of the conceptual model.
- 8) Supplementary materials contain the original video files and complete sequences.

Reviewer #1 (Remarks to the Author):

Review of "Gigantic Jets reveal a stepping process in streamers in the stratosphere" by van der Velde, Montanyà, and López

This manuscript reports an observational study of gigantic jets by using a high-speed camera in conjunction with a sensitive but slower camera. Four GJs were observed on two nights in 2017, and two of them are discussed in detail in the manuscript. It is concluded that the leaders of the GJs reach only 21 km altitude, and above that altitude the GJ discharge develops in a stepwise manner similar to the "pilot

system" propagation mode in negative streamers.

As far as I know, high-speed images of gigantic jets have never been reported. From this point of view, the results reported in the manuscript might deserve to be published in a high-profile journal such as Nature Communications. However, I do not think that the conclusions drawn in the manuscript are fully substantiated by the results presented, which is explained in detail below.

General Comments

1. After carefully reading the manuscript, it appears that the conclusion that the leader discharge only reaches 21 km altitude is drawn based only on the images showing that the GJ channel at this altitude is thinner and brighter. Given that the events occurred at a distance about 350 km from the observation site, I do not think this argument is convincing enough. It is entirely possible that the brightness of a single leader is under the detection limit from this distance in the humid air condition of tropics. The leader tip stops propagating upward after exiting the cloud top is also inconsistent with previous GJ observations [e.g., Pasko et al., 2002; Su et al., 2003; Soula et al., 2011; Liu et al., 2015]. In addition, there may be other reasons why the brightness of the channel section just above the cloud lasts longer. For example, the channel just below and above that section has more branches.

1a) The lower section is thin and brighter. Exactly this section is reilluminated as the "Final Leader" after the Trailing Jet stage. This is not only the case in GJ3 and GJ4 in this paper, but also in various other gigantic jets.

1b) Atmospheric extinction of light works in favor of this section <23 km being a leader. The lower the elevation angle, the more exponential the extinction. Therefore, the lowest section really must be much higher brightness.

1c) About the presumed inconsistency with other works that the leader tip stops propagating upwards: Rioussset et al. 2010 assessed the leader vs streamer sections in the event of Pasko et al. 2002 exactly the same as us. Su et al. 2003 always spoke of streamers and reported speeds were around $1e6$ m/s. Soula et al. 2011 show a case with a lasting/reilluminating final leader like our GJ events. Also they show filaments connecting to the main channel in their fifth event. This means the main channel has differently charged sections, which is impossible for a leader. Events were blue below 40 km. Furthermore the width of the filaments is increasing with time, from less than 100 m to over 1000 m, which means that no cover layer exists to prevent thermal and electrostatic expansion, contrary to leaders in the lower troposphere.

Lu et al. 2011 clearly show the cloud leader transition to streamers. Liu et al. 2015 (incl. Supplementary movies) despite what they claim as leader altitude, all show a leader just above the visible cloud, and it is the longest lasting part of the entire jet, and flickering with the rest of the flash. They used the speed argument, but there are no morphological indications for a change from leader to streamer at the altitude pointed at.

Color photographs in Lyons et al. 2003 and Wescott et al. 2001, as well as recent color photos (included as Reviewer Files) show obvious color and brightness differences between leader and streamer below 25 km.

1d) Brightness of channel because of other reasons like branching: Not a strong argument. This is seen in almost every GJ at the same altitude range and corresponds with the final leader visible after the Trailing Jet stage in a good number of events.

We now discuss in the Introduction evidence for the leader vs streamers in previous works and arguments as to why one cannot reliably distinguish leaders and streamers based on speed only.

2. As suggested by Raizer et al. [2006, 2007], in order for the discharge channel to maintain its conductivity on the GJ timescale of a few milliseconds to tens of milliseconds, the discharge channel needs to have a high temperature. The cold streamer channel cannot sustain its conductivity too long to continue the upward development of the GJ.

2a) The high-speed images show that this is in fact the case. The luminosity of any feature in the leading jet stage last shorter than 2 ms (although lasting weak glow may not show up in high-speed images). On the other hand, the ambient dielectric relaxation time scale (Pasko et al. 1998) is on the order of 100 ms, so the ionized channels remain. There is no need for a perfectly conducting channel all the way from cloud to ionosphere, as the presence of strong fields drive the electron avalanches at streamer tips. Even so, Luque and Gordillo 2012 suggest that associative detachment of O⁻ ions provide longer lasting conductivity at altitudes above 15 km at fields below breakdown.

Detailed theoretical and modeling studies [Riousset et al. 2010; da Silva and Pasko, 2012, 2013] also indicate that the streamer-to-leader transition can and should occur beyond 21 km altitude for the timescale of GJs. The authors need to address if there are only streamers above 21 km altitude, why the GJ can last that long.

2b) Da Silva and Pasko also show (Fig. 15b) that for a leader of 0.5-2 A, the terminal leader altitude is close to 20 km. So, there is no disagreement with the modeling studies indicated. Rioussset et al. 2010 also indicated the leader altitude below 20 km. The GJ takes long (150 ms) to complete because streamer relaxation and pilot formation apparently take place at time scales of ~10 ms. Subsequently the Trailing Jet stage lasts long because a high electric field persists between the cloud and the lower ionosphere, allowing continued streamer discharges.

3. In the simulations presented in the manuscript, the potential of the leader tip is more than 150 MV. At such a high potential, it is hard to understand why the leader does not continue propagating upward but just stays there.

3a) The question why the leader does not reach higher than 21 km is answered in Da Silva and Pasko 2013: it corresponds to their simulation of a leader of 1 A (in their Fig. 15), which lacks the heating to continue.

Like us, Raizer and Da Silva & Pasko papers then agree that a very large potential is needed for the streamers to reach the ionosphere from a low altitude.

Conversely, it would be hard to understand why a leader with a small potential of 10-20 MV should easily reach 40 km altitude. Such potential would be very common, and yet we do not see gigantic jets commonly. In fact, even leaders reaching just 20-30 km are not observed, which would be possible with even smaller potentials. So we consider the suggestion that a leader with a couple tens of MV already can reach claimed altitudes highly unlikely.

Note that a current of 1 A seems low, but it is really the current density that creates the heating that makes the leader propagate. From Da Silva and Pasko 2013 JGR Fig. 14b it can be seen that the current needed for a certain leader speed varies widely, depending on leader radius. So the leader may as well be wider than the suggested laboratory diameter with tens of A current to reach the same speed, e.g. $5e4-1e5$ m/s, reducing speed above e.g. 20 km (Fig. 15b) and stopping e.g. at 23 km (by the end of the GJ as we observe).

For the stepping model, if the field above the leader tip in the end is maintained at the positive streamer propagation threshold value, the question is again why the discharge plasma does not quickly decay due to attachment and recombination.

3b) A fully conductive plasma does not need to be maintained across the entire jet. The streamers only have to adapt the Ecr+ potential gradient across their previous section in order to continue a next

negative streamer step. These steps apparently happen at ~10 ms intervals (at 30-40 km). The dielectric relaxation time scale resulting from the ambient conductivity is on the order of 100-1000 ms (Pasko et al. 1998) at 30 km. This means while current may not last long in a segment, the ionization stays around, and current retraces the existing channels upon making connection with the ionosphere. Note that several works (e.g. Luque and Gordillo 2012) suggest that associative detachment of O⁻ is an important source of conductivity in low pressure discharges at time scales of milliseconds.

Detailed Comments

L16-19: See my general comments. The stepping feature observed may be just associated with the propagation of negative stepped leaders.

L16) We have added discussion in the paper about the leader vs streamer features with references to literature, and the evidence all points at streamers at this altitude. In our particular case, why would streamers (final jump) above 40 km be suddenly much brighter than leader steps at 35 km? The leader ought to be a hot channel formed by the energy of many streamers. It just does not match.

Also, if the lower half of the GJ would be a leader, it should remain conductive thanks to its high temperature. As we now point out in the paper, upon the FDJ, there is no brightening of the lower jet (as in cloud to ground leaders upon contact with the ground). Not even the stem of the final streamer development brightens. The TJ develops after a delay of several milliseconds. Therefore, the lower half of the GJ is not conductive. The upper parts are, and this is how the ionosphere potential descends into the streamers only down to the altitude we see later as the top of the TJ. Had the lower half been conductive as a leader, there would be no reason for the TJ developing up to that altitude.

L19-20: This is a bit speculative. More detailed comments are given below.

L42-44: Jets can reach this altitude range without developing into GJs. This statement is not correct.

L42) We were only rephrasing the Raizer & Da Silva and Pasko explanation. Of course, the growth to the ionosphere can happen only when a sufficient potential can be provided. The text has been improved.

L61-62: A 0.25 km uncertainty is very small. The GJ channel above the

cloud may not be at the same horizontal location as the associated lightning, and even the detection of the associated lightning may have a large location uncertainty.

L61) We now include a figure with the great circle path vs detected lightning sources and have updated the error margin.

□L63: List an altitude scale on Figure 1.

L63) All figures have now altitude scales added.

□L72-77: Light is severely attenuated by the high humidity and haze conditions in tropics. The events occurred very far away from the observation site, and it is possible that a stepped leader at such a large distance is below the detection limit of the camera.

L72a) We agree that this should have an effect on the registered brightness with elevation angle.

But in that case, how can it be explained that a leader, supposedly emitting brighter and in a more continuous spectrum than streamers, appears darker at 38 km than the streamers at 42 km (GJ4 high speed video)? There is no huge transmission difference to be expected for 4.5 vs 5 degrees elevation. Also, the observed difference between the brighter channel below 23 km and the weaker section above it further supports the thesis that this bright section is the leader, since it has been attenuated more by atmosphere and yet it still is brighter. Our new Figure 1 color scale allows to see this more clearly than our previous figure.

The long-lasting pixel may indicate that there are more leader branches below and above this cloud exit point. I am not convinced that the leader only reaches about 21 km altitude.

L72b) The point of this was that there is a clear transition just above the visible cloud between a section that remains brighter and narrow (1x16 pixels!), and filaments which are less bright and laterally expanding. This lowest, brighter section is also the only part flickering at the end, suggesting it is directly in connection with the cloud lightning. This is seen also very clearly in the Supplementary movies of Liu et al. 2015 just above the visible cloud.

L77-80: How do you know the tropopause height is 16 km and the temperature was -77 C? Was it from a balloon? If so, provide the balloon sounding, where it was launched from, and the time of the launch. Also, the coldest satellite cloud top temperatures were about -85 C for the June 30 event, and about -78 C for the August 14 events (see attached images). Thus, the cloud top was at/above the tropopause for all events.

L77) The tropopause height comes from balloon soundings and NARR, between 12Z the day before and 12Z after. The soundings were launched quite far from the site, but they are very similar and rarely show much difference in tropopause height over time.

The 14 August cloud top temperature of -78 C translates to an altitude of 15 km which is 1 km below tropopause and the NARR sounding has the EL at 250 hPa only (> -45 C) so compared to that, the overshoot is still significant. As the satellite image shows, convection was very weak in comparison with the typical situation. The 30 July storm is stronger, with 16 km top, 16.8 km tropopause and EL near 15 km (in NARR at -72 C), similar, so a higher overshooting top.

We have added more information in the text about the 14 August case, which is the one of the jets studied here in detail.

L84: Give what UTC times correspond to the panels in Figure 2a, b. You have the 0 mark for milliseconds on the x axis, but what UTC time does this correspond to for both GJs? Also, put both the detailed evolution of the GJs (the high-speed images) in the Supplementary file.

L84) This figure was replaced. UTC times are mentioned up to the second. Timing relied on network time protocol, which is usually accurate to 0.05 - 0.3 second.

L89-90: Consider to add a time series of the whole GJs from start to finish for the high resolution (low speed) camera, especially since the fast camera doesn't catch much luminous activity before the FDJ stage.

L89) We have added all movies/sequences to Supplementary Information.

□L92-94: I don't understand the statement. How is the average speed limited by the long exposure time of the slow camera? Is it based on the assumptions that the discharge is stepping? Do you mean the average stepping speed? This is a typical average speed for leaders, but 0.4×10^5 m/s is too small for streamers.

L92) In the exposure time of a slow camera, e.g. 20 ms, if a discharge expanded 2000 m the average speed was 1×10^5 m/s. However, the propagation may not be continuous. A 1000 fps camera may observe 1000 m expansion in the first 1 ms, then no progress for 9 ms, then another 1000 m in 1 ms and no more progress for the remaining 9 ms. The true speed was then 1×10^6 m/s or greater (sub-millisecond step duration), which is not too small for streamers.

Besides, a slow apparent streamer speed can arise when there is a slow leader (not visible yet) coming up where we can already see its streamer zone tip. The corona streamers themselves are fast and may regenerate all the time from the leader tip, but the streamer zone is limited to a distance (1-3 km) from the leader tip. Therefore, streamer tips can be visible advancing with the speed of the still invisible leader.

□L98-101: First, those segments could be leader channels; Second, because the discharge is barely detectable, this speed may not correspond to the upward extension of the channel but fast luminosity variation of existing channels. Third, it is not "almost two orders of magnitude" but slightly more than an order of magnitude.

L98) As for the speed, $2 - 5 \text{ E6 m/s}$ versus $0.5 - 2 \text{ E5 m/s}$ is between 1 and 2 orders of magnitude. 1E5 to 3E6 is 1.48 orders (logarithmic scale). We have already provided arguments why these segments are not leader steps or brightness variations in a leader.

L141-144: High-speed images of downward stepped leaders of CGs show the luminosity of a negative stepped leader is non-uniform along the channel, and also varies over time. The text here is inaccurate.

L141) The reviewer's comment seems to refer to a different part than lines 141-144? We have changed large parts of the text.

□L144-146: Well, light from the channel at a lower altitude may be scattered and absorbed more severely because of the slanted path passing through a longer distance in denser air.

L144) As mentioned before, between 4.5 and 5 degrees elevation (38-42 km) the effect of transmission should be much smaller than the observed large brightness difference between what would be a leader vs streamer zone, so it cannot explain it. It is confirmed by the trailing jet later that there is no dramatic absorption of light.

□L146-156: First, the duration of the plasma created by streamers should be checked to see if a pilot node can exist that long. Second, as mentioned above, the higher observed speed may not correspond to the upward propagation of a discharge channel.

L146a) The fact is we see steps, at the measured intervals, so there must be a space stem/pilot.

The dielectric relaxation time is 100s of ms, the 3-body attachment time scale 2-20 ms at 32.5-40.5 km. So this seems fine, although at lower altitudes it is tens of microseconds. But in a streamer zone

conductivity can be maintained by overlapping streamers at different initiation times, and by retrograde positive streamers. Besides that, according to Luque and Gordillo (2012), electrons are detached from O- even at fields below the breakdown threshold, and this can provide lasting conductivity.

□Stepping Model: This model may be overly simplified to describe GJs. It is basically a static model. The authors need to first validate the model by comparing some common parameters that are reported by previous modeling studies of GJs, leaders, or streamers. Second, the timescales of the process involved should be discussed and should match the timescale and speed of GJs.

L146b) The model is very similar to those in Raizer (2006, 2007) and Da Silva and Pasko 2013 (GRL). It calculates a decrease of potential over the streamer zone with distance from the leader tip, using the streamer stability (critical) fields. The only difference is their use of a scaling height (7.2 km) compared to our MSIS-90-E atmospheric density values. This still leads to similar results for potential needed from a leader at given altitudes.

We then incorporated the Gallimberti mechanism for pilot propagation in negative streamers. We only wish to qualitatively explain the observed structure of the GJ. Time scales would indeed be good to include, but this seems complex and a full model is beyond our objective.

L168: For negative leaders, U_0 has a negative value. Do you mean $|U_0|$? This should be clearly indicated.

L168) Yes. It has been corrected.

□L179: For such a large potential, why does the leader tip not continue propagating upward? This needs to be explained.

L179) See our answer above under 3a)

□L191: 'but not below (for reasons explained before)' --- what are these reasons? Because of attenuated light? Please make this clearer here.

L191) The section has been rewritten and this part was removed.

□L194-195: These potentials are very large and may be unphysical. Additional explanation is required to show that those potentials are realistic for leaders.

OV: We added the requested explanation. Note that we are talking about very rare events which require extraordinary circumstances. If a typical charge distribution could produce GJs, we would see them everywhere.

It is unlikely that a similar storm has ever been sampled by balloon electric field soundings. Most such measurements come from New Mexico mountain thunderstorms, not from the tropics where moisture content is much higher (and possibly the resulting charge density). We show that if we increase Krehbiel et al. (2008) negative charge center by 50% in dimensions (7.5 x 7.5 km x 3 km) and 50% in charge density (2.25 nC/m³), cloud potential can quickly go up to over 600 MV instead of 100-150 MV.

□L218-222: See my general comments above. To conclude that the gigantic jets are streamers, those three comments need to be clearly addressed. The last sentence of this paragraph also somewhat contradicts with this conclusion because the leader during LJ stage is not bright enough to be seen and it is entirely possible that the leader reached higher altitude.

L218a) We believe that we have reasonable answers for these comments. During the LJ stage the leader likely is still invisible, and makes a visible step to 21.7 km during/after the FDJ, as shown in Figure 1 (it can be zoomed in to see it better).

To definitively say where the ascending negative leader stops and the streamers dominate requires more evidence than a long lasting pixel above the cloud during the FDJ stage.

L218b) This was a section of 1 pixel wide and 16 pixels tall in GJ3. (a leader channel is predicted to be sub-pixel diameter). The same section is standing out after the trailing jet stage. This feature is seen in many GJs (see PDF with over 20 images of different cases now provided as Files for Reviewers)

Conversely, Liu et al. 2015 who claim the leader reaches altitudes like 42 km do not show any morphological differences at given altitudes to distinguish leader from streamers. However, like cases shown in the PDF, morphological evidence is present in all their movies that the leader stays at 18-23 km (is significantly brighter and always the last section to stay luminous and flickering after the TJ).

L227-223: Production of X-rays and Gamma-rays by leaders or streamers doesn't necessarily lead to TGFs. This is too speculative.

L227) We need very large fields in front of a leader tip to get runaway electrons beaming upwards (high leader potential and upward directed field for electrons). We made a short comment at the end of the paper about that, which we think is relevant given the very large potential arising from our analysis.

Reviewer #2 (Remarks to the Author):

Review of the manuscript titled "Gigantic Jets reveal a stepping process in streamers in the stratosphere" submitted for publication in Nature Communications by van der Velde et al.

The reviewed manuscript presents for the first time gigantic jet (GJ) observations with 1 ms temporal resolution. The authors use the combined information from two cameras, one with high temporal, and another with high spatial resolution, to infer the propagation mode of the discharges in the stratosphere. The key conclusion of the paper is that GJs are mostly streamers, and a lightning leader channel only reaches up to ~20 km altitude.

Although the observations are very interesting, the authors make a lot of conclusions in the paper that are not supported by the images shown. The key issue here is the argumentation that GJs are mostly streamers just because they don't look like lightning leaders in the troposphere. This is a tricky statement to be made.

We have now included discussion about the features of leaders vs streamers and previously reported GJs in the Introduction. Note that images we provided as Suppl. Materials included a photograph clearly showing a purple shaft coming from the cloud top, with an embedded white leader channel is so much narrower and still looks like a lightning channel as in the troposphere. Scaling laws suggest the radius of a leader channel is $0.3 \text{ mm } N/N_0$: about 5 mm. They are clearly wider or widening, which suggests the absence of the ion cover (Bazelyan and Raizer, 2000) that leaders have.

Other factors need to be considered, such as the electrodynamics involved and the plasma composition. These factors can only be probed with electromagnetic field measurements and with spectroscopy, which are not available in the present investigation. Moreover, the numerical model presented does not seem to be physically correct. The issue may only be the lack of proper description. But with the (little) information given in the paper, this reviewer can only conclude that the model is wrong.

My overall impression is that the interpretation is not supported by the available experimental data and by the simulation performed. There are a lot of conclusions that cannot be seen from the four figures in the paper. For all these reasons, I am inclined to reject this paper. But at this time I will recommend the paper to be returned to authors for major revision. The paper may be suitable for publication if it is reframed as "the first high speed observations of GJs" and more careful conclusions are made about the discharge physics.

We reassessed all aspects of our data and of previously reported GJs. Our conclusion that the GJ consists of streamers above about 20 km is consistent with previous observational and modelling studies, as we explain in the new manuscript.

Based on previous work of the same authors, I was also expecting to see a more illustrative analysis of the meteorological scenario involved in the production of these GJs.

This is outside the scope of the paper, but we have added a figure with satellite images. A paper focused on meteorological conditions is soon to be submitted to a journal.

Below you may find additional items to be considered:

1. Line 1: Title. "Gigantic Jets reveal a stepping process in streamers in the stratosphere". The terminology streamer stepping means something else in laboratory discharges. You can see streamer stepping in a single streamer channel, where subsequent discharges reuse the same channel and propagate further than the 1st one. What you see in the images is not streamer stepping, but the complicated dynamics in a streamer zone of a lightning leader. Negative lightning leaders likely step, i.e., pause for some time before propagating again, as a result of this complicated dynamics.

1) We have now modified the title. We agree the GJ is a streamer zone of a leader. The leader tip itself we show to remain at lower altitudes.

2. Lines 15-16: "... At a temporal resolution of 1 ms ..." The authors should clarify here that the key conclusions of the paper do not come from the camera with 1 ms temporal resolution. The authors state that the initial propagation is the in the streamer and not leader mode. But that conclusion comes from the lower temporal resolution camera. The high-speed camera doesn't even see the initial propagation.

2) The conclusion about the leader altitude indeed comes from the other camera, combined with the same features being visible in previously reported GJs. We have now described an additional conclusion from the high speed camera (besides that the discharge proceeds in steps): The lower half of the jet and even the stem of the final jump streamer plume is not re-illuminated as a conductive channel would upon reach the other electrode. This means that section is not conductive, and therefore it is not a leader. This is also why the Trailing Jet develops to the altitude it does: only the streamers in the upper part of the GJ were conductive enough to adopt the ionosphere potential. Therefore a long-lasting field remains, which drives the TJ.

3. Lines 41-44: "The decrease of air density ... jump to the ionosphere." Please add reference to this statement.

3) It has been added.

4. Figure 2: I'm assuming that the images shown in panel (2) are from the fast camera, which has 1.1 ms temporal resolution. Where are the missing frames? They need to be shown here to justify the conclusions. Is there anything present in those frames?

4) Those images contain no visible features (save for -11.1 ms in GJ-3 which is afterglow of the development in -12.2 ms), and were removed to save lots of space. However, we have replaced the figure with one without gaps. Additionally, the full footage will be included as supplementary material.

5. Figure 2: The dotted arrows in panel (2), I believe, indicate the propagation of the discharge. Is the reader expected to see the bidirectional propagation described in the text from this figure? That is not possible. There is no clear bidirectional propagation in this figure.

5) The luminosity is extending downward, it can mean two things: a) an overall luminosity increase extending the visible section; b) bidirectional growth (downward), however it does not continue downward, so it probably is not. However, we replaced the figure.

6. Figure 2: Features that are only slightly above the noise threshold of the camera need to be taken with a grain of salt. Please do not base the main conclusions of the paper on such features.

6) The only feature close to the visual detection limit among the noise is the first step at 32.2 km altitude in GJ3. But we confirmed that by leaving this frame out of a stack, the channel is not reproduced so well as by including it, suggesting it is indeed a real feature.

7. Line 127: "After this darker period, beadlike structures ..." This statement is very interesting. However, it is not supported by Figure 2. The formation and propagation of the described bead-like structures needs to be shown in the imagery.

7) We have improved it in the new Figure 3 that replaces Figure 2.

8. Lines 139-156: The entire Interpretation section needs to be revised. The key idea that an inferred stepwise propagation is associated with the formation of a pilot system reinforces the idea

that a lightning leader channel is involved and not the other way around. I would not expect leaders in the stratosphere to look like the ones in the troposphere.

7) The lightning leader is involved by its electric potential, just does not propagate high. Multiple space stems and pilot systems can occur in its streamer zone (e.g. Gallimberti et al. 2002; Petersen et al. 2008). These space stems do not evolve into a bidirectional space leader as they do in lightning.

There is enough photographic evidence that shows leaders coexist with streamers in the stratosphere, the leaders do look exactly like those near the ground, while the streamers are blue/purple and seem to be bundled, with characteristic branching angles from the main channel, reconnections to the main channel, etc. We include images as Reviewer files.

The key characteristic of leaders is that they must be hot and conductive with minimal potential gradient and are predicted to be very narrow even at stratospheric altitudes. Their ion cover prevents lateral expansion. They cannot bundle tightly together or produce side branches that connect again to the main channel (as for example GJ 5 in Soula et al. 2011).

9. Lines 158-211: The entire Stepping Model section needs to be revised. It is very difficult to connect Figures 3 and 4 with the Gallimberti [2002] model which the authors say that they are using (which anyways is an obsolete framework).

9a) This is a very simple electrostatic representation following Raizer et al. (2006; 2007) and Da Silva and Pasko (2013a). So, potential drops with distance across the streamer zone according to the stability (or critical) field value (E_{cr-}) which is scaled with density. Gallimberti et al. (2002) comes in when the streamer zone tip potential reaches the ambient one, and it cannot propagate further. From that point, the potential of the streamer tip is relaxed to the potential of the curve defined here by E_{cr+} , the stability field of positive streamers which are typically observed in the laboratory (Les Renardières Group, 1981). These retrograde positive streamers are confirmed also in the "ember" secondary TLE discharge included in this version of the manuscript.

Editorial Note: The figure below is reproduced from Gallimberti et al. *Fundamental processes in long air gap discharges. Comptes Rendus Physique* 2002;3(10):1335-1359. Copyright © 2002 Académie des sciences/Éditions scientifiques et médicales, published by Elsevier Masson SAS. All rights reserved.

We include here the Fig. 14 (left) of Gallimberti et al. 2002 rotated and mirrored to match the leader at the bottom right and streamer zone slope corresponding to the lines in our Figure 5. Note that the ambient potential slope beyond the leader tip is not displayed in our case, because it drops very fast to small values. When the negative streamer tip potential reaches the ambient potential, we assume a sort of relaxation process (Gallimberti et al, 2002) brings the potential back up to renew the field in front. This allows new forward negative streamer development. The weaker slope corresponds likely to that of positive streamers retracing the old negative streamer section. These positive streamers, as well as detached electrons from O- (Luque and Gordillo 2012), serve to bring more electrons to the streamer zone tip, enhancing its potential.

But if you look at Figure 14 (left) of Gallimberti [2002] you can see that after the emission of the streamer corona flash from the leader tip, the potential in the streamer zone rises, so that a lower electric field E_{cr} exists in the streamer zone.

9b) We improved the schematic figure in Figure 5a to include this in-between stage of Figure 14 (left) of Gallimberti et al. (2002) for the first step.

After relaxation and emission of a bidirectional streamer corona, the potential rises even further. Nothing of this sort is seen in Figures 3 and 4.

9c) This is not correct, it was displayed. The potential rises further to the weaker slope, corresponding to the critical field for positive streamers which come out backwards from the pilot. We have now labeled these slopes and processes in Figure 5a and its caption.

The authors need to describe their modeling in more detail, maybe in the Supporting Information manuscript. Perhaps showing profiles at different time instants illustrating the different stages of the bidirectional corona development during one of the steps.

9d) We hope to have described it more clearly now. The code for the conceptual model is supplied. Note that intermediate relaxation is not modeled, only the resulting "reset" of potential to that of the Ecr+ slope.

10. Lines 158-211: In the early models, the need for having a leader propagating up to some altitude in the stratosphere appeared in order to allow for GJs to be formed from realistic voltage drops. In the proposed model, the authors find that 300 MV is required when adopting realistic values of Ecr. This should be seen as a red flag for the proposed simulation methodology. What is the maximum realistic value for the potential difference between charge layers in an entire thunderstorm?

10) The potential is consistent with simulations in Raizer, Da Silva and Pasko. If you start lower, the potential needs to be higher. This is no different with our stepping version.

Note that the Krehbiel/Riousset models generally use conservative charge (density) configurations with values close to those measured in thunderstorms - those are likely to occur. This may not be the case for GJ storms, which are very rare, and have not been sampled by electric field soundings.

We would rather consider it a red flag if the required potential is small, as it can be produced by many thunderstorms, yet we do not see upward leaders and gigantic jets everywhere. Note that increasing by 50% the negative charge region dimensions and charge density brings the potential into range of what is needed for the GJ to form with a leader reaching to 18-21 km. We include discussion about this in the paper.

11. The two montages included as supporting information (SI) need to be included/described in the SI manuscript.

11) We include it.

We thank the reviewers for their comments.

[redacted]

Reviewers' comments:

Reviewer #1 (Remarks to the Author):

I have carefully read the revised manuscript and authors' response. The authors have made great efforts in revising the manuscript, and the compilation of the images from recent GJ observations is excellent. However, most of my previous comments are not addressed and there are still major issues in authors' interpretations of the observations. I don't think the manuscripts is acceptable for publication in Nature Communications.

Comments

1. The authors' conclusion of the leaders in GJs reach a maximum altitude of about 20 km is based on two facts: the lasting luminosity of the GJ channel around this altitude and the color images of GJs. Regarding the lasting luminosity of the GJ channel base, we can consider the high speed imaging of lightning leaders or return strokes. High speed images show that the luminosity of a leader is not uniform along its channel and its luminosity doesn't fade away uniformly. As an example, there are typically 1-2 leader channels that are illuminated by the return stroke and many preceding leader channels are not illuminated. During the decay of the luminosity of the illuminated channel, some sections of the channel last much longer than the other sections. Therefore, the lasting luminosity is not a defining factor of a leader channel. For the color images of GJs, it is commonly accepted that when the temperature is reached about 1500-2000 K, streamer-to-leader transition occurs. This temperature is not very different from streamer channel temperature and it is not expected that its spectrum is very different from the spectrum of a streamer channel. In fact, there are a couple of examples of space leaders in Edens et al. [2014] which appear similar to streamers in color.

2. The suggested leader current of 0.5-2 A cannot explain the charge moment change of at least a few hundred C-km observed during the upward propagation of GJs [e.g., Cummer et al., 2009; Liu et al., 2015]. In addition, if a leader with a potential of over 1 MV could only produce such a small current, there would be problems in explaining leader currents of hundreds of amperes in lightning.

3. "Conversely, it would be hard to understand why a leader with a small potential of 10-20 MV should easily reach 40 km altitude." I am not sure why this question is raised by the authors. After going through the references, the authors may be talking about the paper by Liu et al. [2015]. In that paper, the suggested 10-20 MV is the leader tip potential when it reaches 40 km altitude. It is not the leader potential when the leader just exited the cloud.

4. Why are GJs so rare? This is an open question, and there can be many contributing factors, including a large charge imbalance required, successful escaping of leaders, and upward propagation of the escaped leaders. A combination of these factors can make formation of GJs difficult.

5. Regarding the conductivity of the channel. A conductive channel is required to deliver a large fraction of the high potential at the origin of the leader to the leader tip. "The high-speed images show that this is in fact the case. The luminosity of any feature in the leading jet stage last shorter than 2 ms ..." A dark channel doesn't mean that it is not conductive. There are many examples: dark

sprite streamer channel from high-speed images, the faded return stroke channel before the dart leader of the next return stroke, etc. I don't follow the argument that the 100 ms dielectric relaxation time helps sustain the channel conductivity. The dielectric relaxation time is not the timescale for the duration of conductivity. For the role of the detachment of O⁻ ions to sustain the conductivity of the channel [Luque and Gordillo, 2012], that study only includes electron-ion chemistry that is relevant to the altitudes and timescales of sprites. Three-body processes that play a more important role at lower altitudes are ignored. The conclusions from that paper cannot be directly applied to lower altitudes without careful checking.

6. Past studies suggest that there can be as many as one million streamers in the streamer zone of a leader tip. It is hard to believe that the few, bright branches near the base of GJs are individual streamers. Streamers appear brighter at high altitudes can be understood by similarity laws. The total amount of charge in the streamer head is inversely proportional to neutral density, so is the photon emission rate.

7. For the width of the channel, the channel can appear wider because of luminosity saturation and light scattering. In addition, there is no obvious physical mechanism to explain the expansion of a formed streamer channel.

8. Regarding the conclusion of stepwise propagation shown by high-speed images. The quality of the images is not good enough to unambiguously show the stepwise propagation. The luminosity of the GJ discharge during its development can vary, particularly because the GJ discharge is connected to in-cloud lightning activity and the luminosity can be affected by the in-cloud discharge activity.

Reviewer #2 (Remarks to the Author):

Second round of review of the manuscript titled "Stepwise propagation observed in Gigantic Jets" (new title) submitted for publication in Nature Communications by van der Velde et al.

The authors have made substantial effort to address this reviewer's concerns. The manuscript contains new figures and they are described differently. However, I still keep my opinion that the figures do not support the extensive list of conclusions. I am also disappointed that the most important conclusions do not come from the high-speed camera observations (i.e., are not seen by the high-speed camera). The bidirectional propagation of the pilot system of streamers is not seen in the GJ images. Only, perhaps, in the ember event.

For this version of revision/corrections, I will focus my comments on the modeling section of the paper. This section is weak and possibly incorrect. I have examined in detail the provided source code. Now I understand why Figure 5b (new number) doesn't look like the cited reference [Gallimberti et al., 2002, Figure 14a]. Figure 5b does not show the ambient potential, before the establishment of the streamer zone, like Gallimberti does. The Figure 5 caption is not very descriptive.

The key result from Figure 5b is that the jump altitude of a gigantic jet is determined from the stability field for positive streamer propagation. This is remarkably the assumption used in the cited theoretical references, but not acknowledged by the authors, i.e., 18.6 km altitude is the jump altitude of a GJ with 277 MV leader potential, according to the references in the paper. The threshold for negative streamer propagation only determines the step sizes. The paper is actually in line with previous theoretical efforts, and not showing evidence that they are incorrect, as the authors convey.

The similarity between these results and the cited theoretical references would be evident if the authors added a Methods section describing the calculations performed in the simulation code (with references to the proper equations). The model is highly-inspired by the cited references, with the only quantitative change being the MSIS profile, which does not add any new physics, as perhaps the authors conveyed.

The electric field profile shown in Figure 5c makes no sense at all. Note that the potential distribution in Figure 5b was obtained by integrating an electric profile (not shown in the paper, but evident from the code). Therefore, if you start by prescribing the electric field profile, how can you get a different electric field profile in the end (shown in Figure 5c)? In some regions the electric field is substantially above the breakdown threshold. Could you comment on the consequences of it?

The last sentence of the paper regarding TGFs is speculative at this point, but could be connected to the paper if an actual electric field profile is calculated (instead of the incorrect one shown in Figure 5c).

The authors use the argument of rarity to justify the high leader potential required to launch the GJ streamers to the ionosphere. This argument could be better supported by a meteorological analysis of the charge structure in the convective system analyzed. Not done by the authors.

Dear Editor and Reviewers,

Thank you for the opportunity to improve our manuscript.

We hereby present the revised text, new figures and associated files. The manuscript has been almost completely rewritten to accommodate the observations of our Oct-Nov 2018 campaign in Colombia, which resulted in three new gigantic jet recordings with the faster intensified high speed camera we relocated from Curaçao. Two of them are discussed in the new manuscript, recorded at 5000 images per second, one of them is precision-timed and compared to ELF radio signatures from Cape Verde and Duke University, as well as with GLM luminosity (satellite lightning mapper). We count at least 11 previously undocumented features of the jet evolution observed by the high speed camera.

The introduction has been rewritten to provide an overview of interpretations of the jet nature in earlier papers. The event locations, ember discharge discussion and the electrostatic model have been moved to supplementary information, so the main paper now focuses on the high speed gigantic jet analysis. We have rewritten some of the argumentation that reviewers found problematic.

We would like to urge the reviewers to comment exactly on lines in our manuscript instead of general comments to our responses. Note how we attempted to provide a comprehensive discussion of all new aspects of the observed gigantic jets, and we believe our discussion and conclusions remain very close to the features discovered by use of the high speed camera for the first time since 2002.

Best regards,
Oscar van der Velde

Reviewer #1:

Review of "Stepwise propagation observed in Gigantic Jets" by van der Velde, Montanyà, and López

I have carefully read the revised manuscript and authors' response. The authors have made great efforts in revising the manuscript, and the compilation of the images from recent GJ observations is excellent.

Thank you. We have now added two new gigantic jets at 5000 images per second, allowing to see more detail.

However, most of my previous comments are not addressed and there are still major issues in authors' interpretations of the observations. I don't think the manuscripts is acceptable for publication in Nature Communications.

Comments

1. The authors' conclusion of the leaders in GJs reach a maximum altitude of about 20 km is based on two facts: the lasting luminosity of the GJ channel around this altitude and the color images of GJs.

This is correct, but additionally the brightness of this channel is much higher, especially when considering that the light traverses a more dense atmosphere at low elevations. This narrow bright channel is a feature in most GJs where the lowest elevations near the cloud top can be observed. It is featured very prominently again in a new paper by Boggs et al. (2019), where due to short distance unfortunately the earlier frames are overexposed. It is observed in GJ 4 to grow vertically (see figure in supp. Information and in the image series of Pasko et al. 2002), and also has been pointed out by Rioussat et al. 2010 as where streamers-to-leader transition occurs. This section also directly flickers in synchronization with the in-cloud lightning flash as seen in Liu et al. 2015 supp. materials. So, in contrast to the lower half of the jet, this channel section has all the behavior and properties we know of lightning.

Regarding the lasting luminosity of the GJ channel base, we can consider the high speed imaging of lightning leaders or return strokes. High speed images show that the luminosity of a leader is not uniform along its channel and its luminosity doesn't fade away uniformly. As an example, there are typically 1-2 leader channels that are illuminated by the return stroke and many preceding leader channels are not illuminated. During the decay of the luminosity of the illuminated channel, some sections of the channel last much longer than the other sections. Therefore, the lasting luminosity is not a defining factor of a leader channel.

In the case of the GJ, whenever the clouds or horizon permit, images show a very sharp drop in brightness with altitude, in the 18-23 km range (see references in text). This is not a random altitude and does not have anything to do with fading of channels, or branching, or final altitude reached: look at Lu et al 2011 or Liu et al. 2015 where this bright section is visible when the discharge is growing (no decay) and also straight without any branching at all (referring to an earlier comment of the reviewer that a leader section would be brighter because a branched structure above).

The leader lasts longest where the temperature was highest and/or where electric current remained. Even if the 20-40 km altitude section is a leader, its temperature is apparently much lower than the section below 20 km. That has not been explained before by papers proposing the leader to 40 km, and we find it more consistent with observations to explain the 20-40 km section being a form of streamer corona from the leader tip at 20 km. This is totally plausible in the light of figure 2d of Da Silva and Pasko 2013a.

Gradients of luminosity in leaders generally take place over much longer existing channel sections, while stronger gradients are observed directly behind negative leader tips in highspeed images (not in front). The comparison with a return stroke does not apply to the GJ, which is emitted without cloud-to-ground return stroke happening, nor cloud to ionosphere stroke (the current density during the brightest part is 6 orders of magnitude below that of a leader, let alone a return stroke, see values Da Silva and Pasko 2013ab and our discussion of ELF data).

For the color images of GJs, it is commonly accepted that when the temperature is reached about 1500-2000 K, streamer-to-leader transition occurs. This temperature is not very different from streamer channel temperature and it is not expected that its spectrum is very different from the spectrum of a streamer channel. In fact, there are a couple of examples of space leaders in Edens et al. [2014] which appear similar to streamers in color.

We agree the brightness and color of space leaders (or space stems – before heating) can be in the range between streamers and leaders. The Edens segment did not extend and connect so it was not a space leader but a space stem. However, there is another aspect of morphology: once the streamer-to-leader transition occurred, the leader channel is very narrow and much brighter than the streamer zone. It is clearly notable in Edens et al. 2014.

2. The suggested leader current of 0.5-2 A cannot explain the charge moment change of at least a few hundred C·km observed during the upward propagation of GJs [e.g., Cummer et al., 2009; Liu et al., 2015]. In addition, if a leader with a potential of over 1 MV could only produce such a small current, there would be problems in explaining leader currents of hundreds of amperes in lightning.

The CMC measured during our GJ 12 is 25 C·km during the LJ, excluding the final jump of another 40 C·km. The current moment is about 2 kA km during LJ. Results in 100 A, consistent with mentioned studies. However, the 0.5-2 A value comes from the model scenario of Da Silva and Pasko 2013b fig. 15 which results in a terminal leader altitude and speed around 20-22 km as we claim. Note that Da Silva and Pasko 2013 emphasize that the heating to propagate the leader tip is performed by current density in A/m², not current. They used a laboratory leader radius of 0.3 mm $1/e$ N₀/N. But they actually argue why in the atmosphere the radius is expected to be larger. So, keeping the same current density, a 3-4x wider channel easily has a current of 100 A. We describe this in the manuscript. It must be noted that leader widths have never been determined at these altitudes.

3. “Conversely, it would be hard to understand why a leader with a small potential of 10-20 MV should easily reach 40 km altitude.” I am not sure why this question is raised by the authors. After going through the references, the authors may be talking about the paper by Liu et al. [2015]. In that paper, the suggested 10-20 MV is the leader tip potential when it reaches 40 km altitude. It is not the leader potential when the leader just exited the cloud.

This potential is just inferred for that altitude using the Raizer/Da Silva and Pasko electrostatic formulation. Since a leader is very conductive, the potential drop over it is minimal, so if it has 20 MV at 40 km, it may already have had a similar potential exiting the cloud top. Since this is a normal value, such leaders would be seen regularly, but are not.

In that paper, I am not sure how they arrived to the values of e.g. 70 MV when the leader exited the cloud, though.

Note that we modified the discussion and this section is no longer there.

4. Why are GJs so rare? This is an open question, and there can be many contributing factors, including a large charge imbalance required, successful escaping of leaders, and upward propagation of the escaped leaders. A combination of these factors can make formation of GJs difficult.

We agree. However, our point was that storms with “normal” properties may not produce GJs and we may find they are found in a long tail of a statistical variations of charge distributions.

5. Regarding the conductivity of the channel. A conductive channel is required to deliver a large fraction of the high potential at the origin of the leader to the leader tip.

Yes, but a slightly conductive channel like a streamer corona will also bring up a smaller fraction of the original potential.

“The high-speed images show that this is in fact the case. The luminosity of any feature in the leading jet stage last shorter than 2 ms ...” A dark channel doesn’t mean that it is not conductive. There are many examples: dark sprite streamer channel from high-speed images, the faded return stroke channel before the dart leader of the next return stroke, etc.

The reviewer is right, however, the very slow response to events like reaching ionosphere or fast intracloud pulses suggest the possibility the lower jet is not so conductive. It re-ionizes. The interstroke interval of tens of ms in a CG flash cools the channel, so it also definitely lost a lot of its conductivity, and the dart leader as result is a slower process than the return stroke wave.

I don’t follow the argument that the 100 ms dielectric relaxation time helps sustain the channel conductivity. The dielectric relaxation time is not the timescale for the duration of conductivity. For the role of the detachment of O⁻ ions to sustain the conductivity of the channel [Luque and Gordillo, 2012], that study only includes electron-ion chemistry that is relevant to the altitudes and timescales of sprites. Three-body processes that play a more important role at lower altitudes are ignored. The conclusions from that paper cannot be directly applied to lower altitudes without careful checking.

It does not sustain channel conductivity. It does provide a partially ionized pathway favored by new developments during the course of the jet. What sustains the discharge is the potential transferred across the streamer corona. It is commonly known that corona remains active for as long as its driving electric field. It is not the duration of a single streamer that matters.

We have simplified the discussion in supp. info to not speculate about O⁻ detachment.

6. Past studies suggest that there can be as many as one million streamers in the streamer zone of a leader tip. It is hard to believe that the few, bright branches near the base of GJs are individual streamers. Streamers appear brighter at high altitudes can be understood by similarity laws. The total amount of charge in the streamer head is inversely proportional to neutral density, so is the photon emission rate.

This has been extensively documented by laboratory evidence of negative streamers (see publications by Les Renardières Group, or Bazelyan and Raizer 2000, Chapter 2) and by Edens et al. 2014, which the reviewer just mentioned. The reviewer acknowledged that the brightness and color of what Edens et al indicated as a “space leader” is not much different from the streamers – thereby accepting the paradigm. As you get closer to a leader tip, the number and rate of streamers and its spatial density increases greatly.

If we take the reviewer’s comment to imply that the much lower brightness, blue filaments also are leaders, please explain where are the streamers and why suddenly at 40 km altitude they would appear and much brighter than the “leader” below. There should always be a streamer zone in front of the leader tip, not only during the final jump.

7. For the width of the channel, the channel can appear wider because of luminosity saturation and light scattering. In addition, there is no obvious physical mechanism to explain the expansion of a formed streamer channel.

This is clearly not the case, as there is no saturation in no part of the jet in our high resolution images. As for light scattering: the image is capable of showing a leader channel at the bottom of 1 pixel wide without any glare around it, so this also does not apply.

Streamer lateral expansion may not be documented widely, but clearly happens in event nr. 5 of Soula et al. 2011, where a side streamer of the jet bends back towards the main channel and reconnects, something leaders never do, not even after a return stroke. Furthermore, if streamers can get warm as well, as the reviewer just suggested, they should expand. Another way of expanding is continued streamer activity through the corona channel, which may not go through the exact same trajectories as previous streamers. You can see this in GJ 2 (supp.info) where a corner gets smoothed.

8. Regarding the conclusion of stepwise propagation shown by high-speed images. The quality of the images is not good enough to unambiguously show the stepwise propagation. The luminosity of the GJ discharge during its development can vary, particularly because the GJ discharge is connected to in-cloud lightning activity and the luminosity can be affected by the in-cloud discharge activity.

We have added two new GJ sequences recorded at 5000 fps. In one of them (GJ 12), a step is very clearly observed again, and it lasts 3 frames (0.5 ms), which is remarkable, because it means it is not a forward stroke like we see in stepped leaders. If a space leader, it would grow backward, connect and produce a stroke, but does not. This step was not accompanied by a cloud flash. Also, these GJs show downward propagation at the start of the final jump, indicating a bidirectional process (like in the ember event in supp. info), which was suspected for GJ 4 but not clear enough. We added time-altitude-luminosity plots showing also the steps.

Reviewer #2 (Remarks to the Author):

Second round of review of the manuscript titled "Stepwise propagation observed in Gigantic Jets" (new title) submitted for publication in Nature Communications by van der Velde et al.

The authors have made substantial effort to address this reviewer's concerns. The manuscript contains new figures and they are described differently. However, I still keep my opinion that the figures do not

support the extensive list of conclusions. I am also disappointed that the most important conclusions do not come from the high-speed camera observations (i.e., are not seen by the high-speed camera). The bidirectional propagation of the pilot system of streamers is not seen in the GJ images. Only, perhaps, in the ember event.

Our new manuscript includes two additional jets from our 2018 campaign, and we focused the paper on describing the processes observed. We attribute 11 findings to the high speed camera footage which have never been described before. There is now more evidence that at least the final jump onset involves a downward component. Also, a step was resolved by 3 frames (0.5 ms) in our newly obtained material.

For this version of revision/corrections, I will focus my comments on the modeling section of the paper. This section is weak and possibly incorrect. I have examined in detail the provided source code. Now I understand why Figure 5b (new number) doesn't look like the cited reference [Gallimberti et al., 2002, Figure 14a]. Figure 5b does not show the ambient potential, before the establishment of the streamer zone, like Gallimberti does. The Figure 5 caption is not very descriptive.

The description was in the text. The ambient potential drops very fast away from the leader tip at the spatial scale we are looking at for the GJ (kilometers instead of meters). We have relocated this section to the Supplementary Information.

The key result from Figure 5b is that the jump altitude of a gigantic jet is determined from the stability field for positive streamer propagation. This is remarkably the assumption used in the cited theoretical references, but not acknowledged by the authors, i.e., 18.6 km altitude is the jump altitude of a GJ with 277 MV leader potential, according to the references in the paper.

It was acknowledged in line 269 in the previous manuscript: *“Like previous models, we consider the potential drop with distance across the streamer zone from a leader tip.”*

Note that these values result from the calculation specific to our GJ 3 event, resulting from step size fitting (lines 293-298 in the old manuscript).

Additionally, the cited models would use only the negative stability field for negative gigantic jets, which results in even higher potential needed given a certain leader tip altitude. And the altitude of the final jump is in our presented model determined by the last step that permanently maintains sufficient potential difference to reach the ionosphere. In previous models, this had to be the case right from the leader tip.

Note this part has been moved from the main text to the Supplementary Material.

The threshold for negative streamer propagation only determines the step sizes. The paper is actually in line with previous theoretical efforts, and not showing evidence that they are incorrect, as the authors convey.

We thank the reviewer for his comment. In fact, we did not intend to convey the models we cited are incorrect. We combined the principles of these works.

The similarity between these results and the cited theoretical references would be evident if the authors added a Methods section describing the calculations performed in the simulation code (with references to the proper equations). The model is highly-inspired by the cited references, with the only quantitative change being the MSIS profile, which does not add any new physics, as perhaps the authors conveyed.

We agree the model is inspired by the cited electrostatic models, as we have described. We adopted concepts of Gallimberti et al. with those in Raizer et al. and Da Silva and Pasko and applied it to the case of a negative stepping discharge.

All the script does is already described in the corresponding text. We will just add comments in the script which part corresponds to the Raizer/DSP translation to potential and which to the step reset of Gallimberti et al.

Note this part has been moved from the main text to the Supplementary Material.

The electric field profile shown in Figure 5c makes no sense at all. Note that the potential distribution in Figure 5b was obtained by integrating an electric profile (not shown in the paper, but evident from the code). Therefore, if you start by prescribing the electric field profile, how can you get a different electric field profile in the end (shown in Figure 5c)?

Thanks for the comment. The text in line 264-266 (old manuscript) described the line in the code that explains the internal profile of potential along the discharge (Figure 5b):

“The original leader potential drops across the streamer zone according to the internal electric field E_{cr} scaling with the ratio of atmospheric density at that altitude compared to sea level ($\rho_0 = 1.225 \text{ kg m}^{-3}$)”.

Code: $U_{zn}(i) = U_{zn}(i-1) - z_intv \cdot E_{crn} \cdot (NN0(i+z0./z_intv-1) + NN0(i+z0./z_intv))/2$
(where i points to the present altitude)

However, for Figure 5c, we referred to the *external* field in the space between the corona tip and ionosphere at a given tip position. It was described in lines 309-311: *“Finally, an electric field can be assumed in front of the streamer tips, scaling inversely with the square of the distance from a spherical charge with the potential of the tip (Figure 5c). The charge is assumed to scale directly with the potential.”*

In some regions the electric field is substantially above the breakdown threshold. Could you comment on the consequences of it?

Lines 314-318 (old manuscript) commented: *“This may explain why the discharge no longer pauses above 40 km in GJ3, increases in brightness and develops a large number of branches above 55 km as the field goes above the breakdown threshold and no longer decreases, multiplying the free electrons. The acceleration at fields above E_k is consistent with modeled streamers (e.g. Luque and Ebert, 2010).”* So, surpassing the breakdown threshold is how we explain the branched upper GJ morphology.

The last sentence of the paper regarding TGFs is speculative at this point, but could be connected to the paper if an actual electric field profile is calculated (instead of the incorrect one shown in Figure 5c).

We removed this sentence. We moved this discussion to Supplementary Materials and improved this part by including additional references about TGF and potentials in storms.

The authors use the argument of rarity to justify the high leader potential required to launch the GJ streamers to the ionosphere. This argument could be better supported by a meteorological analysis of the charge structure in the convective system analyzed. Not done by the authors.

Such analysis is out of scope, and needs data of lightning mapping arrays, multiple balloon soundings, and even so, one cannot determine cloud charge directly to obtain the potential of the leader.

Furthermore, there is nothing wrong with an argument of rarity. It is all about the statistical chance of encountering a storm charge distribution deviating from commonly accepted values. Note that we have rewritten this part and is now located in Supplementary Materials.

REVIEWERS' COMMENTS:

Reviewer #1 (Remarks to the Author):

I appreciate the authors' effort to rewrite the manuscript. The decision to move the modeling section to Suppl. Material is also very smart. I think the manuscript has been significantly improved and meets the requirements to publish in Nature Communications.

Reviewer #3 (Remarks to the Author):

Review of Evolution of Gigantic Jets at high imaging rates

By van der Veldt, Montanya, Lopez and Cummer

Overview: This paper documents measurements of Gigantic Jets recorded on the North coast of Columbia during the summer of 2017 and the fall of 2018 using ground based optical observations recorded in at 900 and 5000 images per second. Additionally the data are compared with ELF magnetic field measurements made at Cape Verde and Duke University as well as the Geostationary Lightning Mapper.

This paper is well written, and documents new observations with increased temporal resolution of Gigantic Jets. As pointed out in the paper, Gigantic Jets are one of the few instances where charge movement in the troposphere is directly connected with the ionosphere. As such, this paper clearly meets the barrier for publication in Nature Communications.

The authors have three main sections in the paper documenting the measurements, and a supplementary section containing additional detail of the observations and a model used to help interpret the measurements. While some readers may disagree with the methods used in the model, the manuscript clearly delineates observations from interpretation.

I recommend publication of the manuscript as is, and provide a few comments for the authors to consider before they provide the final version of the manuscript for publication.

General Comments:

In the main portion of the paper why are the methods (lines 397-443) at the end? Would it not read easier if they were moved to before observations (line 76)?

Given that these are the first high speed observations of gigantic jets I urge the authors to provide the raw cine files containing the Phantom data as well as the digital images of the slow data. I imagine they will prove useful in future modelling studies.

Specific comments:

Line 270. Can you estimate the uncertainty in the 100 A in the Leading Jet? I assume it is a combination of the uncertainty in the magnetic field measurement and the range? Similar comment on line 271 and 275.

Line 385. Text states 11 interesting new observations. Lines 286 to 298 document 12 itemized "main observations." Which is it? Clean up the text here.

Line 401. The decay time of the P43 phosphor is approximately 1 ms. Document the decay time, this may prove important in future analysis of the data.

Line 402. Document the number of pixels used in the Miro 3 recording. Suggest giving the approximate spatial resolution of the image.

Line 407 Field of view of the Navitar and associated spatial resolution?

Line 411 and 412. Field of view and number of pixels of the Phantom 7.3? Document the decay time of the P-24 phosphor.

Supplementary Information: This portion of the manuscript provides further examples of the data and a model used to interpret the results.

General Comments: What important implications are obtained in section 2 Electrostatic model? Approximately 100 lines of the manuscript are devoted to the model, but it is not clear what the major results of this section are. Summarize importance at either the beginning or the end of this section.

Specific comments:

Figure S1. Indicate the field of view in degrees in the caption.

Figures S3 and S7. Indicate the altitude and time as in Figure 3 of the main paper. If the axes are the same as in Figure 3 simply state this.

Responses to the comments of Reviewer #3

General Comments:

In the main portion of the paper why are the methods (lines 397-443) at the end? Would it not read easier if they were moved to before observations (line 76)?

This section contains more details than are necessary for the understanding of the experiment and follows after the main text, in line with the style of the journal.

Given that these are the first high speed observations of gigantic jets I urge the authors to provide the raw cine files containing the Phantom data as well as the digital images of the slow data. I imagine they will prove useful in future modelling studies.

We have now uploaded the cine and avi files to the Zenodo scientific data repository, indicated in the main text.

Specific comments:

Line 270. Can you estimate the uncertainty in the 100 A in the Leading Jet? I assume it is a combination of the uncertainty in the magnetic field measurement and the range? Similar comment on line 271 and 275.

We have specified the uncertainties now in the Methods section.

Line 385. Text states 11 interesting new observations. Lines 286 to 298 document 12 itemized "main observations." Which is it? Clean up the text here.

The 11 refers to those specifically observed thanks to high-speed camera. The 12th is a feature observed in GJ 4 in the high-speed images but was resolved better in the slow camera images with higher resolution in GJ 3 and 4. We simplified it to 12 new observations in the line mentioned.

Line 401. The decay time of the P43 phosphor is approximately 1 ms. Document the decay time, this may prove important in future analysis of the data.

Added. In all of our GJ cases the phosphor decay time is shorter than the frame interval.

Line 402. Document the number of pixels used in the Miro 3 recording. Suggest giving the approximate spatial resolution of the image.

Added.

Line 407 Field of view of the Navitar and associated spatial resolution?

Added.

Line 411 and 412. Field of view and number of pixels of the Phantom 7.3? Document the decay time of the P-24 phosphor.

Added.

Supplementary Information: This portion of the manuscript provides further examples of the data and a model used to interpret the results.

General Comments: What important implications are obtained in section 2 Electrostatic model?
Approximately 100 lines of the manuscript are devoted to the model, but it is not clear what the major results of this section are. Summarize importance at either the beginning or the end of this section.

We added a summary of the implications at the end.

Specific comments:

Figure S1. Indicate the field of view in degrees in the caption.

Figures S3 and S7. Indicate the altitude and time as in Figure 3 of the main paper. If the axes are the same as in Figure 3 simply state this.

We added and generally improved the figure axes (also in the main paper)

We thank all reviewers for their efforts which improved the manuscript.